# ARTICLES

## OPEN

# Structure of CRL7<sup>FBXW8</sup> reveals coupling with CUL1–RBX1/ROC1 for multi-cullin-RING E3-catalyzed ubiquitin ligation

Linus V. M. Hopf [1], Kheewoong Baek [1], Maren Klügel[1], Susanne von Gronau[1], Yue Xiong[2,3] and Brenda A. Schulman[1]✉

**Most cullin-RING ubiquitin ligases (CRLs) form homologous assemblies between a neddylated cullin-RING catalytic module and a variable substrate-binding receptor (for example, an F-box protein). However, the vertebrate-specific CRL7<sup>FBXW8</sup> is of interest because it eludes existing models, yet its constituent cullin CUL7 and F-box protein FBXW8 are essential for development, and CUL7 mutations cause 3M syndrome. In this study, cryo-EM and biochemical analyses reveal the CRL7<sup>FBXW8</sup> assembly. CUL7's exclusivity for FBXW8 among all F-box proteins is explained by its unique F-box-independent binding mode. In CRL7<sup>FBXW8</sup>, the RBX1 (also known as ROC1) RING domain is constrained in an orientation incompatible with binding E2~NEDD8 or E2~ubiquitin intermediates. Accordingly, purified recombinant CRL7<sup>FBXW8</sup> lacks auto-neddylation and ubiquitination activities. Instead, our data indicate that CRL7 serves as a substrate receptor linked via SKP1–FBXW8 to a neddylated CUL1–RBX1 catalytic module mediating ubiquitination. The structure reveals a distinctive CRL–CRL partnership, and provides a framework for understanding CUL7 assemblies safeguarding human health.**

The evolutionarily conserved multiprotein cullin-RING ligases (CRLs) form the largest superfamily of ubiquitin E3 enzymes. Humans have an estimated 250–300 different CRLs that collectively regulate virtually all eukaryotic cell biological processes[1–3]. At the core of all CRLs is a heterodimeric subcomplex containing a cullin-family protein and a dedicated ≅10 kDa RING-domain-containing partner RBX1, RBX2 (also known as ROC1 and ROC2) or APC11 (ref. [4]). Humans have nine cullins. Structural studies of seven of these in complex with their RING-domain-containing partners (CUL1–RBX1, CUL2–RBX1, CUL3–RBX1, CUL4A–RBX1, CUL4B–RBX1, CUL5–RBX2 and APC2–APC11) revealed how they achieve the two main functions of an E3: binding a substrate and catalyzing its ubiquitination. First, all these cullins are elongated multidomain proteins. One end connects to a variable substrate-binding receptor, either directly or in a multiprotein complex that contains a specific cullin-binding subunit[1–3,5,6]. For example, while CUL3 binds directly to the BTB domain in its partner substrate receptors, CUL1 binds interchangeably to complexes between SKP1 and various substrate receptor F-box proteins[1–3]. APC2 represents an extreme example; it is embedded in the ≅1.3 MDa multiprotein Anaphase-Promoting Complex/Cyclosome (APC/C) that has its own distinct set of substrate receptor subunits[7,8]. Specific CRLs are named by their cullin number (or APC) with the name of the substrate receptor in superscript.

Second, as in most E3 ligases, the RING domains serve as catalytic elements. The best-recognized function of a RING domain is binding an E2~ubiquitin or E2~ubiquitin-like protein (UBL) intermediate in an activated conformation ('~' refers to a transient thioester bond between an enzyme cysteine and the C terminus of ubiquitin or a UBL)[4]. In so doing, the RING domain stimulates ubiquitin or UBL transfer to a nucleophilic acceptor. The RING

domains of RBX1, RBX2 and APC11 all display such activities. RBX1 and RBX2 recruit particular E2 enzymes that catalyze linkage of the UBL NEDD8 to their associated cullin proteins. Subsequently, in the context of a neddylated CRL, they activate ubiquitin transfer from other E2s, or from ARIH-family RBR E3s, to receptor-bound substrates[1–3]. APC2 is not neddylated, but is activated when a substrate receptor binds to the APC/C. APC11's RING domain binds to the E2 enzyme that transfers ubiquitin to APC/C-bound substrates. However, for polyubiquitination, the cullin APC2 binds the E2~ubiquitin intermediate, while APC11's RING domain binds the substrate-linked ubiquitin that it modifies[9]. Cullin-bound RING subunits also bind a variety of regulatory proteins[1–3]. RBX1, RBX2 and APC11 also promote the proper folding of their cullin partner, by the region N-terminal of the RING domain contributing to an intermolecular β-sheet with the cullin's so-called α/β-domain[6].

The structural mechanisms of the two other vertebrate-specific cullin-RING complexes, CUL7–RBX1 and CUL9–RBX1, remain elusive[1–3]. Although little is known of CRL-like functions of CUL9–RBX1, CUL7–RBX1 associates with the F-box protein FBXW8 and SKP1. The CUL7–RBX1–SKP1–FBXW8 complex was termed 'CRL7<sup>FBXW8</sup>' due to its containing subunits homologous to those in CRL1<sup>F-box protein</sup> E3 ligases[10–17].

CRL7<sup>FBXW8</sup> is essential for proper mammalian development. FBXW8 and CUL7 null mouse embryos are severely retarded in their growth[11,18,19]. Most such animals do not survive to adulthood. CUL7 null mice have a more severe phenotype: most succumb during embryogenesis. CRL7<sup>FBXW8</sup> also promotes neuronal Golgi integrity and dendrite morphogenesis[15], and has been implicated in tumorigenesis as well (reviewed in refs. [17] and [20]). CUL7 is cytoplasmic[21] and has several interaction partners besides SKP1–FBXW8[10,12,15,22], but the full suite and stoichiometries of proteins

[1]Department of Molecular Machines and Signaling, Max Planck Institute of Biochemistry, Martinsried, Germany. [2]Department of Biochemistry and Biophysics, Lineberger Comprehensive Cancer Center, University of North Carolina at Chapel Hill, Chapel Hill, NC, USA. [3]Present address: Cullgen Inc., San Diego, CA, USA. ✉e-mail: schulman@biochem.mpg.de

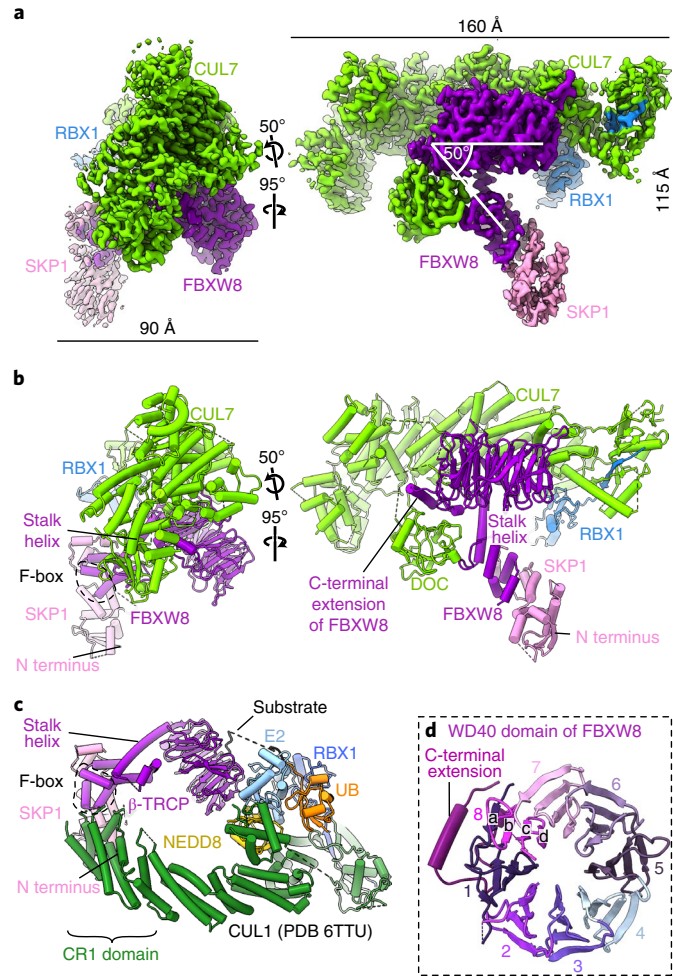

**Fig. 1 | Structure of CRL7^FBXW8. a**, Cryo-EM map of CRL7^FBXW8 (calculated with DeepEMhancer[54]). **b**, Structure of CRL7^FBXW8 in the orientation of the cryo-EM map in **a**. **c**, Structure showing substrate ubiquitination by CRL1^β-TRCP and E2 (PDB 6TTU)[35], with SKP1 aligned in the orientation of CRL7^FBXW8 in **b**. **d**, WD40 domain and C-terminal extension of FBXW8. Blades are labeled and color coded from 1–8 and **a**–**d** strands are labeled on blade 8.

in various assemblies remains unclear. What is clear is that CUL7, RBX1 and CUL9 interact. Also, CUL7–RBX1 binds OBSL1 and CCDC8 in a complex named '3M' in reference to mutations in the *CUL7*, *OBSL1* or *CCDC8* genes causing 3M syndrome. The 3M syndrome is characterized by short stature, and distinctive facial and skeletal abnormalities. The CCDC8 protein was named for its predicted coiled-coil domain, is thought to contain several intrinsically unstructured regions that mediate protein interactions, and localizes at the plasma membrane[15,21–24]. OBSL1 is a putative cytoskeletal adapter protein with multiple immunoglobulin-like domains and a fibronectin type-3 domain[25]. CCDC8 binds to OBSL1, which leads to assembly of CUL7-OBSL1-CCDC8 into a membrane-associated 3M E3 ligase complex and ubiquitination of the plasma membrane cell migration regulatory protein PHLDB2 (also known as LL5β)[26]. In addition, CUL7 and CUL9 also interact with TP53 (refs. [21,27,28]). Notably, CUL7 and/or FBXW8 have been implicated in the ubiquitination of a variety of proteins with diverse roles in cellular signaling and proliferation, including TP53 (refs. [17,21,29]). It is unknown whether CUL7's interaction and functions in partnership with SKP1–FBXW8, with OBSL1-CCDC8, with CUL9, and with TP53 are independent of each other, but the distribution of these proteins

copurifying with each other in various immunoprecipitations suggests that there are multiple distinct CUL7-containing E3 complexes in vivo[26], none of which are understood structurally.

CRL7^FBXW8 might seem to parallel canonical CRLs, in that it comprises both a CUL–RBX subcomplex and a SKP1–F-box protein subcomplex. However, CUL7 is unusually large (≅190 kDa), and the CRL7^FBXW8 complex displays many atypical features. CUL7 has a distinctive ≅100 kDa N-terminal region, homologous only to a similar region in CUL9. This region is predicted to contain armadillo repeats, a ≅15 kDa DOC domain of unknown function and a ≅11 kDa CPH domain (a conserved domain within CUL7, CUL9 and HERC2 proteins) that binds TP53 (ref. [28]). Beyond the structure of the CPH domain and characterization of its interactions with TP53 by NMR[30], little is known of the detailed interactions constituting or made by CRL7^FBXW8. Moreover, features of CRL7^FBXW8 differ substantially from the well-studied CRL1 family wherein SKP1 connects CUL1–RBX1 and an F-box protein. First, whereas CUL1–RBX1 binds nearly 70 different human SKP1–F-box protein modules, CUL7–RBX1 exclusively associates with SKP1 bound to FBXW8 (ref. [10]). Second, the sequence of CUL7 is devoid of the CUL1 domain that binds the SKP1–F-box motif raising the question of how it binds SKP1–FBXW8 (ref. [6]). Third, assays of CUL7 immunoprecipitates from cells cotransfected with wild-type and mutant expression plasmids for CUL7 and TP53 showed CUL7-mediates ubiquitination of TP53 in a manner that, perplexingly, does not require its cullin domain-bound RING-domain partner RBX1 (ref. [21]). Fourth, along the same lines, CUL7 copurifies with NEDD8 but does not appear to be neddylated itself[14,31]. Finally, in vivo, CRL7^FBXW8 curiously associates with CUL1–RBX1. This interaction requires FBXW8: in wild-type mouse embryonic fibroblasts, exogenously expressed Flag-tagged CUL1 immunoprecipitates endogenous CUL7 and Flag-tagged CUL7 immunoprecipitates endogenous CUL1. However, the CUL7-CUL1 interactions were not observed in FBXW8 null cells[18]. Although a number of models have been proposed[18,32,33], it remains unknown how FBXW8 brings together multiple different cullin molecules, and why specifically CUL7 and CUL1. This raises questions of whether and how CUL7 could serve as an E3 ligase. Here, we answer these questions with the cryo-EM structure of CRL7^FBXW8 and structure-based biochemical studies showing its unique architecture and assembly with neddylated CUL1–RBX1 into an active E3 ubiquitin ligase.

## Results

**CRL7^FBXW8 forms a unique cullin-RING ligase assembly.** A cryo-EM reconstruction of a recombinant human CRL7^FBXW8 complex, prepared by coexpressing CUL7, RBX1, SKP1 and FBXW8 in human human embryonic kidney 293S (HEK293S) cells, refined to 2.8 Å resolution overall. The density allowed fitting previous structures of RBX1 and SKP1, and building atomic models of CUL7 and FBXW8 de novo. The final coordinates were generated after multiple cycles of manual rebuilding and refinement (Fig. 1, Table 1, Extended Data Fig. 1 and Extended Data Table 1). Although AlphaFold was released subsequent to structure determination, it is noteworthy that most of the individual domains in CUL7 and FBXW8 superimpose with those predicted with Cα-r.m.s.d. values of <1.3 and <0.8 Å, respectively (Extended Data Fig. 2)[34].

CRL7^FBXW8 adopts a diagonal T-shaped structure. CUL7–RBX1 forms a ≅160 Å long bar that crosses the ≅115 Å long CUL7-FBXW8–SKP1 base of the 'T' at a ≅50° angle (Fig. 1a). The T-junction is formed by five domains from CUL7 wrapping around FBXW8's WD40 β-propeller and stalk helix (Fig. 1b). SKP1's N-terminal region projects away from CUL7–RBX1 and was poorly resolved in the maps, but was visualized at the edge of the complex by docking previous SKP1–F-box protein complex structures into the low-resolution density (Extended Data Fig. 3a). Overall, the structure is well resolved, allowing unambiguous placement of

**Table 1 | Cryo-EM data collection, refinement and validation statistics**

| | CRL7^FBXW8 (EMDB-14547) (PDB 7Z8B) | CRL7^FBXW8–CUL1^NTD (EMDB-14558) |
|---|---|---|
| **Data collection and processing** | | |
| Magnification | 105,000 | 22,000 |
| Voltage (kV) | 300 | 200 |
| Electron exposure (e⁻/Å²) | 60 | 60 |
| Defocus range (µm) | −0.7 to −2.8 | −1.2 to −3.3 |
| Pixel size (Å) | 0.8512 | 1.885 |
| Symmetry imposed | C1 | C1 |
| Initial particle images (no.) | 6,050,000 | 1,377,383 |
| Final particle images (no.) | 355,547 | 363,067 |
| Map resolution (Å) | 2.82 | 5.4 |
| FSC threshold | (0.143) | (0.143) |
| Map resolution range (Å) | 2.6–6.6 | – |
| **Refinement** | | |
| Initial model used (PDB code) | 4P5O, 6O6O | |
| Model resolution (Å) | 2.8 | |
| FSC threshold | (0.143) | |
| Model resolution range (Å) | 2.6–6.6 | |
| Map sharpening B factor (Å²) | −30 | |
| **Model composition** | | |
| Nonhydrogen atoms | 14,463 | |
| Protein residues | 1,896 | |
| Ligands | 3 | |
| **B factors (Å²)** | | |
| Protein | 89.4 | |
| Ligand | 139.9 | |
| **R.m.s. deviations** | | |
| Bond lengths (Å) | 0.004 | |
| Bond angles (°) | 0.529 | |
| **Validation** | | |
| MolProbity score | 1.45 | |
| Clashscore | 8.3 | |
| Poor rotamers (%) | 0.2 | |
| **Ramachandran plot** | | |
| Favored (%) | 98.42 | |
| Allowed (%) | 1.58 | |
| Disallowed (%) | 0 | |

RBX's RING domain and building of residues for most of the protein complex (Extended Data Fig. 3b–e).

The CRL7^FBXW8 T-shape differs from the more ovoid assemblies formed by canonical CRLs (Fig. 1c). CUL7 does not directly bind SKP1. This is explained by CUL7 lacking a cullin-repeat 1 (CR1) domain that is used by cullins 1–5 to bind their cognate substrate receptor modules. Notably, CR1 domain interactions are central to the characteristic architecture of prototypic CUL1-based E3 ligases[6]. CUL1–RBX1 forms half an oval. At one end, CUL1's

CR1 domain and SKP1 curve around to place the F-box protein's substrate-binding domain facing RBX1 bound at the other end of CUL1. The oval is closed when RBX1's RING domain binds a ubiquitin-carrying enzyme to deliver ubiquitin to an F-box protein-bound substrate[35,36] (Fig. 1c). Below, we describe the distinct structural details of CRL7^FBXW8, and how E3 ligase activity can be achieved despite its unique assembly.

**SKP1–FBXW8 resembles canonical SKP1-FBXW assemblies.** Within CRL7^FBXW8, although SKP1 does not contact with CUL7, the SKP1–FBXW8 subcomplex shows all the hallmark features of canonical SKP1 complexes with WD40 domain-containing F-box proteins: (1) the compact three-helix F-box is encased by SKP1's C-terminal helices (Extended Data Fig. 4a); (2) a ≅30 Å long stalk helix rigidly projects from the F-box and (3) the stalk helix continues into a β-propeller (Extended Data Fig. 4b,c). FBXW8's eight-bladed β-propeller overall superimposes well with those of other F-box proteins, most notably human FBXW7 (Protein Data Bank (PDB) 2OVP) and yeast Cdc4p (PDB 1NEX) (Extended Data Fig. 4b)[37,38]. As is typical of WD40 proteins, the β-propeller is closed by the N-terminal strand completing the otherwise C-terminal-most blade (Fig. 1d). However, FBXW8 is unique in that its C-terminal ≅35 residues extend beyond the propeller to form a triangular platform between the propeller and stalk helix. This C-terminal structure makes three-way interactions between FBXW8's first propeller blade and two disparate CUL7 domains (Fig. 1b).

**Unique domains and cullin-RING arrangement in CUL7–RBX1.** The CUL7–RBX1 portion of the structure contains four distinct regions (Fig. 2a,b). CUL7's atypical N-terminal region and C-terminal cullin region bound to RBX1 together form the elongated bar shape that crosses the 'T'. CUL7's N-terminal region comprises a unique collection of domains that cap the N-terminal edge of the cullin region. A small beta-domain (SBD) connects via a poorly resolved 50-residue linker to the first of three short armadillo repeat units (ARM1, ARM2 and ARM3). The ARM3 unit covers the seam between the ARM1 and ARM2 domains, and proceeds into the first helix of the cullin region. The junction between the N- and C-terminal regions is mediated by a 64 Å long bent helix that is both the last helix of the ARM3 unit and the first helix of the cullin region. Altogether, the SBD, ARM1, ARM2 and ARM3 domains pack against each other to wrap around and completely encircle this junction helix. The N-terminal-most domain in the cullin region (the CR2 domain) is embraced on one side by the SBD and on the other by the ARM2 domain (Fig. 2b).

CUL7's cullin region initiates with CR2 and CR3 domains, which are five-helix bundles arranged in tandem, and proceeds with a four-helix bundle (4HB), a mixed α-helix/β-sheet (α/β) region, an α/β-to-WHB linker and the C-terminal WHB domain (Fig. 2c,d and Extended Data Fig. 5a). Individually, most of these CUL7 domains resemble those in other cullins[6] (Extended Data Fig. 5a). However, CUL7's α/β-to-WHB linker differs in such a way that prevents neddylation as described in detail below.

RBX1's N-terminal region forms a β-strand, which is inserted in CUL7's α/β region. Together, these CUL7 and RBX1 elements form an intermolecular cullin/RBX (C/R) domain that requires both proteins for proper folding. However, the relative orientation of RBX1's ensuing RING domain differs from the model arrangement in CUL1–RBX1 (Fig. 2d). RBX1's ten C-terminal residues extend beyond the RING domain, fill a crevice between CUL7's 4HB and WHB domains, and cement a CRL7-specific domain arrangement (Fig. 2e). Although the WHB domain is the C terminus of canonical cullins, CUL7 has an additional 50 residues not overtly visible in the cryo-EM density. It is tempting to speculate that CUL7's C terminus corresponds to patchy cryo-EM density covering RBX1's RING

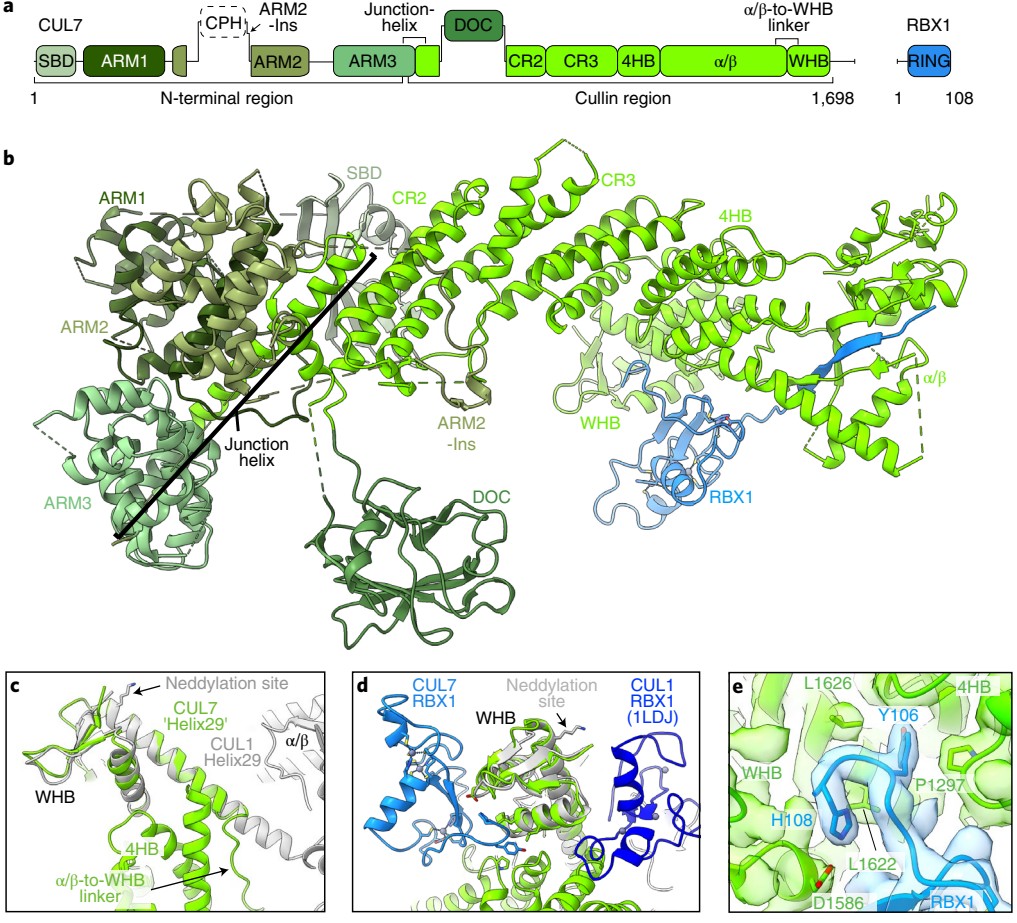

**Fig. 2 | Architecture of CUL7–RBX1. a,** Domains of CUL7 and RBX1. **b,** Structure of CUL7–RBX1 from CRL7[FBXW8] (excluding FBXW8–SKP1 for a clearer view of CUL7), with domains colored according to **a**. **c,** Superposition of CUL7 and CUL1 (PDB 1LDJ)[6] aligned on the WHB domain showing closeup of α/β-to-WHB linker region. **d,** Superposition of CUL7 and CUL1 (PDB 1LDJ) aligned on WHB domain showing relative orientations of RBX1 RING domains in CRL7[FBXW8] and CUL1–RBX1 (PDB 1LDJ)[6]. **e,** C terminus of RBX1 anchored to WHB and 4HB of CUL7 shown in DeepEMhancer map.

domain, although the map does not allow unambiguous attribution of this region (Extended Data Fig. 3b).

The two other structured domains from CUL7—the ARM2-Ins (an extended loop insertion in ARM2) and the DOC domain—both bind FBXW8 (Fig. 3a). The CPH domain is not visible in the cryo-EM maps of CRL7[FBXW8] and is presumably flexibly tethered to the ARM2 domain.

**Unique cullin-F-box protein assembly.** CUL7 binds FBXW8 through an elaborate multipart interface that differs entirely from CUL1 interactions with F-box proteins. First, the concave surface from CUL7's CR2, CR3 and 4HB domains encases roughly half the rim of the top side of FBXW8's β-propeller (Fig. 3a). Grooves between CUL7 CR3 and 4HB domain helices embrace intra- and inter-blade loops from blades 5–8 of FBXW8 (Fig. 3b, c). Extended loops between the central strands of blades 5 and 8 secure the two edges of this interface. Second, CUL7's ARM2-Ins protrudes nearly 40 Å from the ARM2 domain, fills the groove between CUL7's CR2 and CR3 domains, meanders around the crevice between the FBXW8's β-propeller and C-terminal extension, and then returns to complete the ARM2 domain (Fig. 3c). Finally, CUL7's DOC domain nestles in a furrow between the stalk helix and C-terminal extension of FBXW8 (Fig. 3d). The atypical assembly explains why CUL7 uniquely binds FBXW8 but no other F-box proteins: the interactions are with domains, domain insertions and sequences unique to FBXW8, and even those with the WD40 domain involve residues

that are not conserved even with its closest homolog FBXW7 (Extended Data Fig. 5b,c).

The structure also provides a rationale for why our purification of FBXW8 required coexpression not only with SKP1 to bind its F-box, but also CUL7–RBX1 (Methods). The interface is highly hydrophobic; it seems likely that exposure of the hydrophobic surfaces would lead to aggregation of SKP1–FBXW8 purified on its own. In total, 85 residues from CUL7 interact with 91 residues—nearly 20%—from FBXW8 to bury roughly 3,500 Å² of surface area (Extended Data Fig. 5d).

Comparing interface residues between the cryo-EM structure and those predicted using AlphaFold raises the possibility some key interacting regions in CUL7 and FBXW8 may fold and/or rearrange to form the interaction. In particular, the differences from the models suggest the CUL7 ARM2-Ins and DOC domain, and FBXW8 blade 5 and blade 8 loops and the C-terminal extension are considerably remodeled and/or reoriented in the complex (Extended Data Fig. 2b).

**Purified recombinant CRL7[FBXW8] assembly is inactive.** Canonical CRLs exhibit a wide range of E3 ligase activities, in partnership with various NEDD8- and ubiquitin-linked carrying enzymes. These activities depend on multivalent interactions, at minimum involving the RBX RING domain, an E2 and its active site-linked NEDD8 or ubiquitin, and also other contacts, for example between a cullin-linked NEDD8 and a ubiquitin-carrying enzyme[35,36,39].

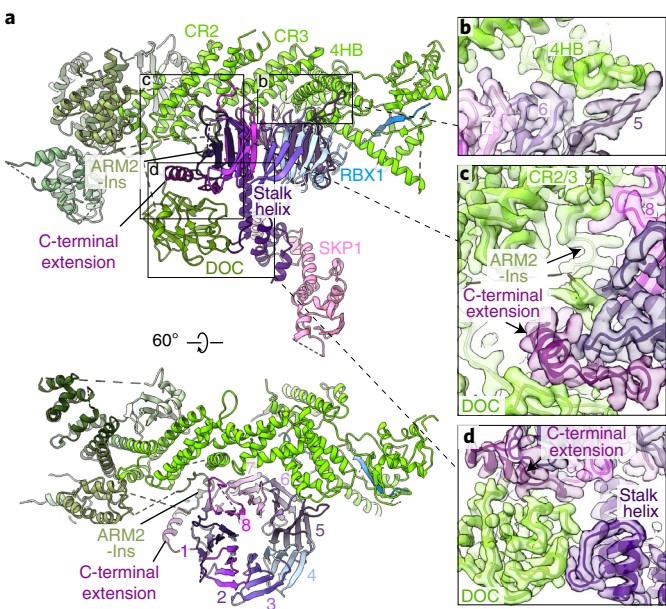

**Fig. 3 | CUL7-FBXW8 interface. a**, Structure of CRL7[FBXW8] with CUL7 domains colored as in Fig. 2 and interfaces between CUL7 and FBXW8 highlighted in squares. FBXW8 is colored in a gradient to visualize the individual blades 1–8, the stalk helix and the C-terminal extension. **b–d**, Close-ups of CUL7-FBXW8 interfaces with corresponding cryo-EM density calculated with DeepEMhancer. **b**, Interface between CUL7 4HB and FBXW8 blades 5 and 6. **c**, Interface between FBXW8 C-terminal extension with CUL7 ARM2-Ins and DOC domain, between CUL7's CR2 and CR3 domains, and an extended loop from FBXW8 blade 8. **d**, Interface between CUL7 DOC domain and FBXW8 stalk helix and C-terminal extension.

The structure of CRL7[FBXW8] is inconsistent with even the most rudimentary of these canonical CRL NEDD8 and ubiquitin E3 ligase activities. A previous structure showed how CUL1 is neddylated on a specific lysine in the WHB domain[40]. Comparing the structure of CRL7[FBXW8] showed two features that prevent it from attaining the arrangement required for neddylation despite CUL7's WHB domain displaying a lysine corresponding to the CUL1 neddylation site. First, a long, rigid 'Helix29'—which is conserved in structures of unneddylated cullins 1–5—acts like a wand to direct the targeted lysine into the active site of the RBX1-E2~NEDD8 intermediate (Fig. 4a)[40]. However, CUL7 lacks this long, rigid Helix29. Instead, the corresponding region forms the α/β-to-WHB linker, an extended amphipathic structure not found in the other cullins (Figs. 2a,c and 4a,b). The hydrophobic face of the α/β-to-WHB linker is embedded in a hydrophobic surface from CUL7's 4HB domain (Fig. 4b,c). These interactions, along with additional contacts between CUL7's WHB domain with the 4HB domain and with RBX1's RING domain, fixes CUL7's WHB domain in an orientation incompatible with neddylation (Fig. 4d–e). Second, previous studies also showed that neddylation requires a specific RBX1 RING-domain orientation. This is required to bind and position the E2~NEDD8 intermediate with its active site adjacent to the cullin's target lysine (K720 in CUL1, Fig. 4d)[41]. However, in CRL7[FBXW8], RBX1's RING domain is secured in an inactive arrangement by intercalation of its C-terminal residues between CUL7's 4HB and WHB domains (Fig. 2e). The position of RBX1's RING domain in CRL7[FBXW8] prevents binding an active E2~NEDD8 intermediate as shown in a model based on the structure visualizing CUL1 neddylation (Fig. 4e). The arrangement in CRL7[FBXW8] also prevents RBX1's RING domain from binding an active E2~ubiquitin intermediate

as shown in a model based on a structure visualizing CRL1 substrate ubiquitination (Extended Data Fig. 6a,b).

E3 ligase activities have not been reconstituted with biochemically pure CRL7[FBXW8]. We therefore tested activity with our purified recombinant components. In canonical CRLs, RBX1 promotes NEDD8 transfer to the cullin's WHB domain lysine from either neddylating E2 UBE2M or UBE2F, while the homologous RBX2 only functions with UBE2F (ref. [42]). Examining neddylation side-by-side with canonical cullin–RBX complexes showed that in control reactions, by the first time-point, fluorescent NEDD8 was fully transferred from UBE2M or UBE2F to CUL1–RBX1, and from UBE2F to CUL5–RBX2. However, CRL7[FBXW8] was not substantially modified by either NEDD8 E2 (Fig. 4f and Extended Data Fig. 6c).

We next probed automodification as a readout for intrinsic ubiquitin E3 activity. Under conditions when the control canonical CUL–RBX1 complexes (wild-type or neddylated) were autoubiquitinated in the presence of E1 (UBA1) and E2 (UBE2D3), CRL7[FBXW8] was relatively inactive, similar to the nonneddylatable CUL1 K720R mutant (Extended Data Fig. 6d). One possibility could be that CRL7[FBXW8] preferentially uses an alternative E2. However, autoubiquitination was also not observed with 26 tested E2s (Extended Data Fig. 6e).

In complexes with neddylated canonical cullins, RBX1's RING domain can also promote ubiquitin chain formation by UBE2G1 and by UBE2R-family E2s[43]. RBX1 associated with unneddylated canonical cullins can also promote such polyubiquitination with UBE2R1 (ref. [44]), which is stimulated by interactions between an acidic C-terminal region of UBE2R1 and a basic canyon on canonical cullins[45] that is notably lacking in CUL7 (Extended Data Fig. 6f). These activities can be assayed even in the absence of a substrate, by monitoring transfer of fluorescent ubiquitin from a preformed E2~ubiquitin intermediate to an unlabeled acceptor ubiquitin (~ here refers to the thioester linkage between E2 catalytic cysteine and ubiquitin C terminus). However, CUL7–RBX1 was inactive under conditions when RBX1 complexes with CUL1 promote di-ubiquitin synthesis (Extended Data Fig. 6g). Taken together, the results are consistent with the structural findings that RBX1's RING domain in CRL7[FBXW8] is hindered from binding an active E2~NEDD8 or E2~ubiquitin intermediate.

**FBXW8–SKP1 bridges CUL7–RBX1 and CUL1–RBX1.** In vivo, CUL7 was found to associate with CUL1 in an FBXW8-dependent manner[18]. The CRL7[FBXW8] structure suggests the mechanism: the canonical CUL1-binding residues in CUL7-bound SKP1 and the F-box are fully exposed and thus could mediate the interaction of CUL7 with CUL1. Comigration in size-exclusion chromatography (SEC) showed CRL7[FBXW8] directly binds full-length CUL1–RBX1 (Fig. 5a), but not an N-terminally truncated version lacking the CR1 domain (Extended Data Fig. 7a). Furthermore, when bound to CRL7[FBXW8], CUL1–RBX1 is excluded from binding another SKP1–F-box protein complex (harboring a monomeric version of the F-box protein β-TRCP1) (Extended Data Fig. 7b–d). Indeed, a cryo-EM map of a complex between CRL7[FBXW8] and the N-terminal region of CUL1 (comprising the CR1, CR2 and CR3 domains), refined to a resolution of 4.6 Å visualized the interactions: SKP1–FBXW8 within the CRL7[FBXW8] complex binds CUL1's CR1 domain in a canonical manner (Fig. 5b).

**CRL7[FBXW8] can recruit a neddylated CRL1 substrate in vitro.** Given that CRL7[FBXW8] lacks E3 activity in our assays, but it binds CUL1–RBX1, we asked if CRL7[FBXW8] could in principle serve in a substrate receptor manner for a neddylated CRL1 complex. If this were the case, then a CRL7[FBXW8]-interacting protein could potentially serve as a substrate in vitro. The only such interaction that has been reconstituted with pure proteins is CUL7's CPH domain binding to TP53. Indeed, we found that CRL7[FBXW8], TP53 and CUL1–RBX1 can form a complex based on comigration by SEC, and this

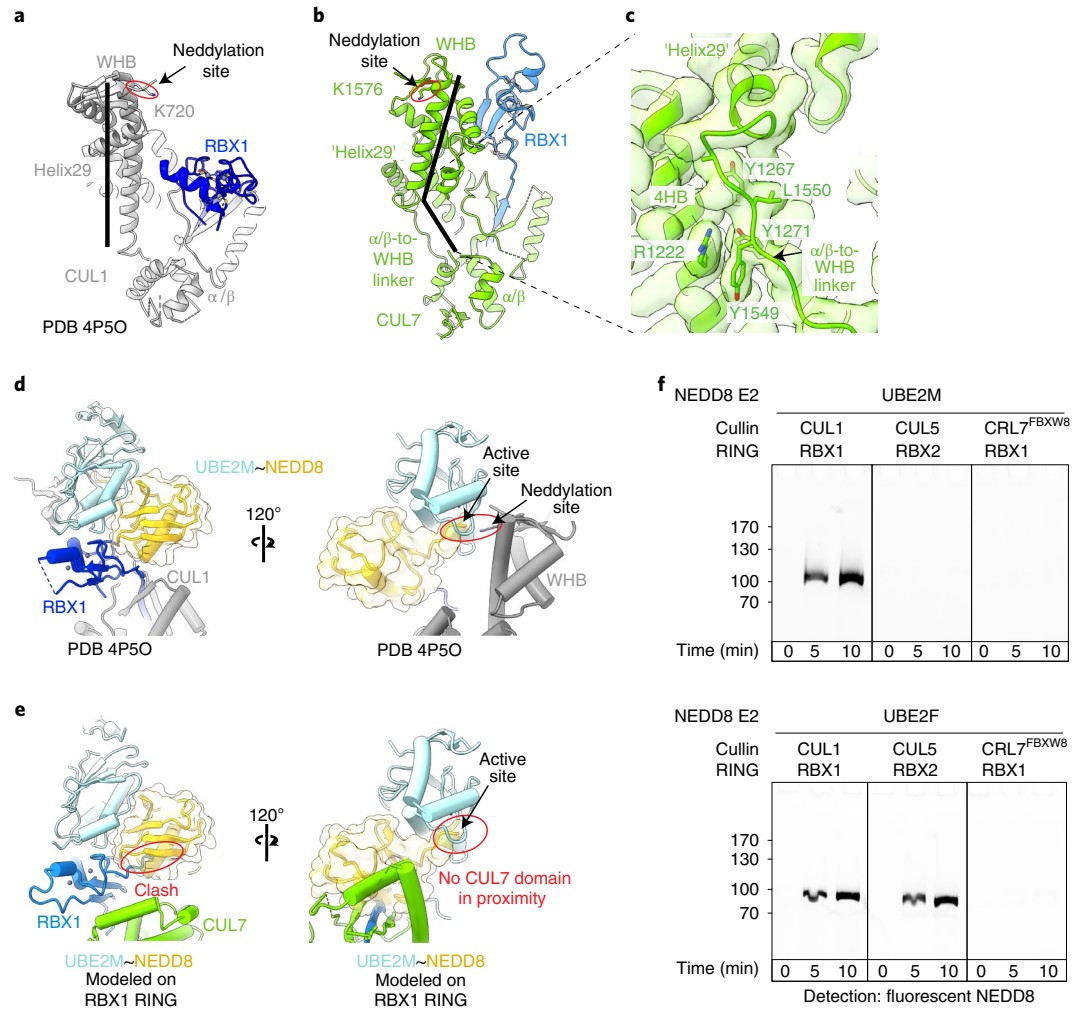

**Fig. 4 | CUL7 is not neddylated under conditions allowing efficient neddylation of canonical cullins. a,b**, Structural arrangement of α/β-to-WHB linker region comparing CUL1 (PDB 4P5O)[40] in **a** with CRL7[FBXW8] in **b**, views are aligned on α/β domain. PDB 4P5O required use of a CUL1 version with the neddylation site K720 mutated to arginine to prevent transfer of NEDD8 and visualize the catalytic architecture. K720 was modeled in the structure here for clarity. **c**, Interface between CUL7's α/β-to-WHB linker region and 4HB shown in cryo-EM map calculated with DeepEMhancer. **d**, Two views of structure visualizing CUL1 neddylation (PDB 4P5O). **e**, Modeling activated RBX1 RING–UBE2M~NEDD8 (PDB 4P5O) shows this clash with CRL7[FBXW8], depicted by aligning RBX1 RING domain from both structures. **f**, In vitro neddylation assays using the indicated cullin-RING complex and NEDD8 E2 (UBE2M or UBE2F), NEDD8 E1 (NAE1-UBA3) and detecting fluorescently labeled NEDD8 in SDS–PAGE gels (samples of both gels derived from the same experiment and gels were processed in parallel, *n* = 2 technically independent experiments).

depends on CUL7's TP53-binding CPH domain (Extended Data Fig. 7e–j). The CPH domain is connected to the ARM2 domain by equal to or roughly 45- and 28-residue long linker sequences, which are presumably flexible given that this domain is not observed in the cryo-EM map. In principle, the CPH domain—which itself spans equal to or roughly 25 Å—could extend over a wide radius relative to the rest of the complex to deliver TP53 to the ubiquitination active site (Extended Data Fig. 8a).

Previously, we found that immunoprecipitated CUL7 was shown to mediate mono- or oligoubiquitination of TP53 (ref. [21]). Notably, a CUL7 mutant unable to bind RBX1 showed similar activity[21]. We were able to reconstitute such TP53 ubiquitination using entirely purified components, dependent on CRL7[FBXW8] and neddylated CUL1–RBX1 (Fig. 5c, lanes 1–4). We thus, used TP53 as model substrate to probe mechanistic implications of the CRL7[FBXW8]–CUL1–RBX1 complex. Several observations are consistent with CUL7 recruiting TP53 for neddylated CUL1–RBX1-mediated ubiquitination. First, ubiquitination depends on CUL7's TP53-binding CPH domain (Fig. 5c, lane 5). Second, neddylated CUL5–RBX2, which

does not bind SKP1–F-box protein substrate receptor modules, cannot substitute for neddylated CUL1–RBX1 (Fig. 5c, lane 6). Finally, the CRL7[FBXW8]-dependent TP53 ubiquitination is decreased with catalytically impaired neddylated CUL1–RBX1 mutants (nonneddylatable CUL1 K720R, Fig. 5c, lane 7), or CUL1–RBX1 modified with Q40E mutant NEDD8 (the Q40E mutation blocks noncovalent interactions between NEDD8 and CUL1 that allosterically activate CUL1)[35] (Fig. 5c, lane 8). Future studies will be required to investigate potential cellular functions of TP53 ubiquitination by the distinctive CRL7[FBXW8]-neddylated CUL1–RBX1 assembly. Nonetheless, the data demonstrate that CRL7[FBXW8] can biochemically act as a CRL1 substrate receptor (Fig. 5b,c and Extended Data Fig. 8a,b).

**3M syndrome mutations.** The physiological importance of CUL7 is underscored by its mutation causing developmental defects resulting in the human hereditary short stature disorder 3M syndrome[23]. *CUL7* mutations account for 70% of 3M syndrome cases, with the remainder caused by mutations in *OBSL1* and *CCDC8* occurring in a mutually exclusively manner between these three genes.

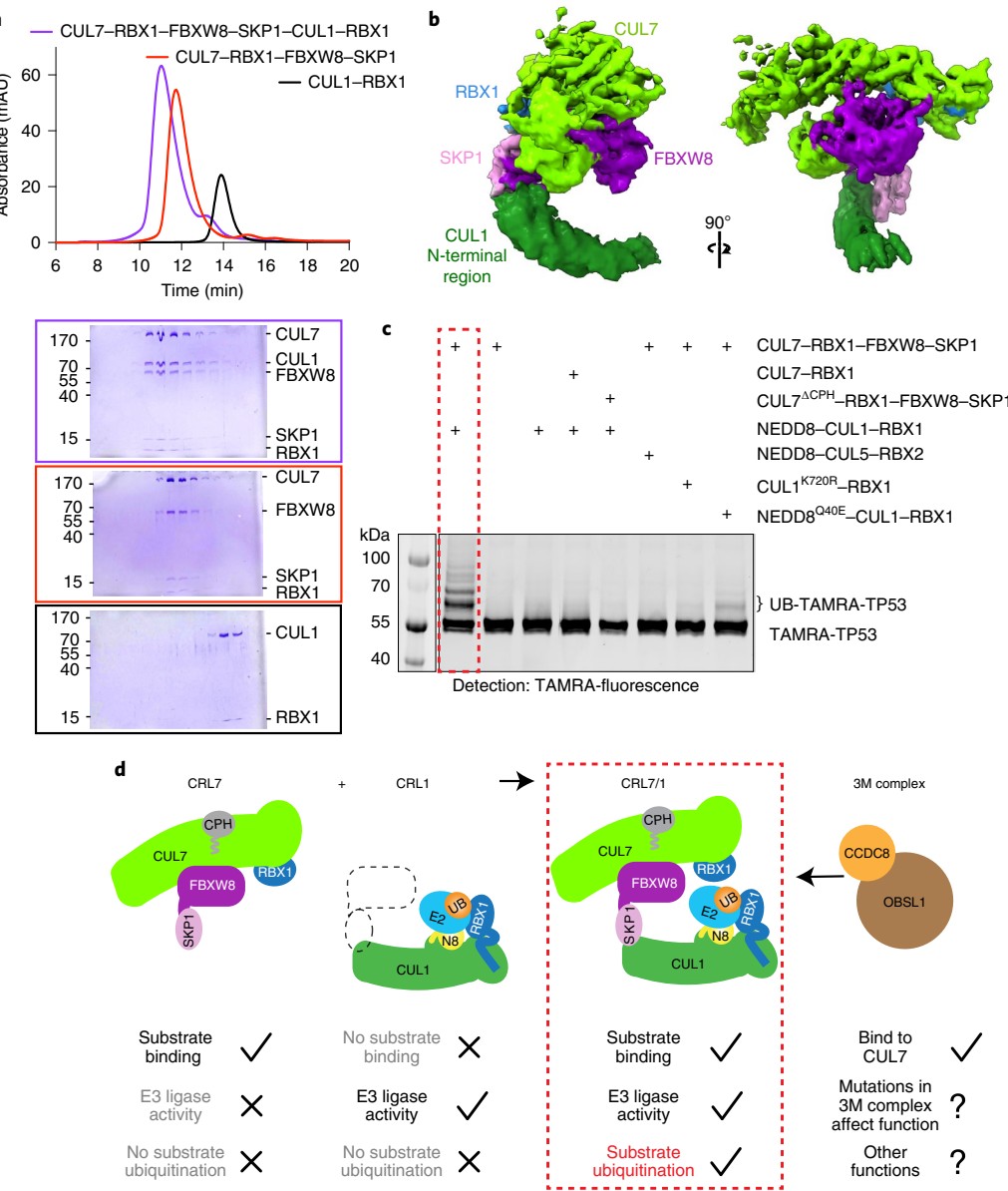

**Fig. 5 | CRL7^FBXW8 forms an active E3 ligase with NEDD8–CUL1–RBX1. a**, SEC experiments comparing migrations of CRL7^FBXW8, CUL1–RBX1 and the mixture of both through a Superose 6, 5/150GL column, visualized by total absorbance (280 nm) and the corresponding elution fractions on Coomassie stained SDS–PAGE gels. **b**, Cryo-EM map of CRL7^FBXW8–CUL1 N-terminal region (residues 1–410) in two orientations. **c**, In vitro assay showing TP53 ubiquitination by CRL7^FBXW8 and neddylated CUL1–RBX1. Assays detected TP53 with TAMRA appended at the N terminus, and contained indicated versions of CUL1, CUL5 and CUL7 complexes, E1 (UBA1), E2 (UBE2D3) and ubiquitin (n = 2 technically independent experiments). **d**, Model of the active E3 ligase formed by interaction between CRL7^FBXW8 and neddylated CUL1–RBX1. N8, NEDD8; UB, Ubiquitin.

CUL7 interacting with OBSL1 and CCDC8 may be compatible with CUL7 incorporation into CRL7^FBXW8 or binding to TP53 (refs. [15,17,21,23,24,46,47]), although it is unclear if such interactions are functionally related to the 3M complex since neither FBXW8 nor TP53 are mutated in the disease (Extended Data Fig. 8c). Nonetheless, the CRL7^FBXW8 assembly allows obtaining insights into potential structural effects of 3M syndrome mutations. Mapping mutations onto the structure shows that most are premature truncations that would affect CUL7 binding to RBX1 and presumably formation of the 3M complex, as this depends on C-terminal regions of CUL7 binding to OBSL1 (ref. [15]). Although missense mutations map throughout the structure, several do map to interprotein interfaces in CRL7^FBXW8, including with FBXW8 (P861S in the DOC domain, and Q1246G in the 4HB) and RBX1 (W1448R, G1452D, H1464P, Q1469R in the

α/β-domain), as well as the interdomain interface between the CR2 and CR3 domains (L1014R) (Extended Data Fig. 8d,e). In agreement with the structure, mutation of H1464 and Q1469 located in CUL7's β-sheet that binds RBX1's N terminus, or the frame-shift at E1594 that truncates the RBX1-binding region of the WHB domain, were shown to eliminate binding to RBX1 (ref. [24]). Thus, although FBXW8 has not been found to be mutated in 3M syndrome patients, the CUL7 mutations at the interface with FBXW8 may suggest a role in the function of the 3M E3 ligase complex.

## Discussion
The structure of CRL7^FBXW8 reveals an atypical assembly between a cullin and F-box protein, and provides rationales for its perplexing features and the curious interaction between CUL7 and CUL1.

CRL7[FBXW8] displays many fundamental differences from canonical CRLs. First, extensive interactions anchor CUL7's WHB domain and RBX1's RING domain in an arrangement that is incompatible with both neddylation and ubiquitination (Figs. 2c–e and 4a–e, and Extended Data Fig. 6a,b). In particular, RBX1's RING domain is uniquely lodged against CUL7 in a way that would prevent binding to an activated E2~NEDD8 or E2~ubiquitin intermediate. Accordingly, our purified CRL7[FBXW8] complex lacks intrinsic NEDD8 or ubiquitin ligase activities in our assays (Fig. 4f and Extended Data Fig. 6c–e), agreeing with published proteomics studies that identified NEDD8 modification of other cullins but not CUL7 (refs. [15,48]). Second, five CUL7 domains directly bind FBXW8 elements not found in other F-box proteins (Fig. 3). This explains CUL7's striking specificity for FBXW8 (refs. [10,15]). Finally, the unique CRL7[FBXW8] assembly exposes the F-box and SKP1. These elements are thus available to bridge CUL7 and CUL1–RBX1 (Fig. 5b and Extended Data Fig. 8a). CRL7[FBXW8] binding to neddylated CUL1 may also solve the riddle of how affinity-tagged NEDD8 copurifies CUL7 despite lack of neddylation of CUL7 itself[14,31].

To address why CRL7[FBXW8] binds another cullin-RING complex, our structural modeling suggested that CRL7[FBXW8] resembles a multiprotein substrate receptor for a neddylated CRL1 complex (Extended Data Fig. 8a). There has long been precedent for CRLs relying on multiprotein complexes to recruit substrates, from one of the first human F-box proteins discovered: the F-box protein SKP2 binds a cyclin–CDK2–CKSHS1 complex to recruit the substrate phosphorylated p27 (ref. [49]). Indeed, we were able to biochemically reconstitute ubiquitination of TP53 recruited by CUL7 (Fig. 5c). Although we were not able to reconstitute ubiquitination of other CRL7[FBXW8] substrates, it seems likely that some substrates may be avidly recruited by interaction with other domains of CUL7 and/or FBXW8 domains, much like multivalent interactions underlie recruitment of substrates to CRL1[SKP2-CKSHS1-Cyclin-CDK2].

CRL7[FBXW8] joins a growing list of E3 ligases that function in partnership with other E3 enzymes. For some E3 pairs that coassemble in E3–E3 complexes—for example, the two RBR-family E3s HOIL-1L and HOIP in the linear ubiquitin assembly complex, and the HECT-type E3 Ufd4 with RING or RING-like E3s Ubr1 and Ufd4, respectively—each E3 mediates sequential steps in polyubiquitination[50,51]. However, each E3 within these complexes is thought to achieve activity by binding an E2~ubiquitin intermediate, whereas we did not observe E3 ligase activity with purified CRL7[FBXW8] (Extended Data Fig. 6d,e). Rather, CRL7[FBXW8] partnering with neddylated CUL1–RBX1 may share some conceptual parallels with canonical neddylated CRLs partnering with ARIH-family RBR E3s such that one E3 (the CRL) recruits substrate while the other (the ARIH-family member) mediates ubiquitination[36,41,52].

At this point, many questions linger about the cellular functions of CRL7[FBXW8] and other complexes formed by CUL7–RBX1. First, can CUL7–RBX1 directly ubiquitinate a substrate without CUL1–RBX1? The identities of direct substrates and functional roles of CRL7[FBXW8] ubiquitination—including for TP53—remain poorly understood. Although deletion of subunits of CRL7[FBXW8] affects the cellular levels of some proteins, it is not clear which of those bind to CRL7[FBXW8] directly, or whether their ubiquitination depends on CRL7[FBXW8] forming a complex with neddylated CUL1–RBX1 (ref. [17]). Second, what is the role of RBX1 in CUL7 complex given that the purified recombinant CRL7[FBXW8] complex appears to be catalytically inactive? From a structural perspective, RBX1 appears to serve as a crucial stabilizing support for CUL7's cullin region (Fig. 2). But does RBX1 play additional functions beyond promoting proper CUL7 folding in the context of other binding partners? For example, in the context of the 3M complex, does RBX1 display NEDD8 or ubiquitin ligase activity? Do currently unknown mechanisms activate neddylation? Are there contexts in which the RING in CUL7–RBX1 is an active E3 ligase? Third, CUL7 associates with many other proteins, including CUL9, which is homologous to CUL7 but with an additional ARIH1-family E3 ligase domain, yet little is known about how and why these proteins interact and mediate regulation[13,14,16]. Is the CUL7–CUL9 complex catalytically active? Fourth, does the CUL7-FBXW8 interaction play a role in preventing 3M disease? Although SKP1–FBXW8 copurifies with the 3M complex from some cell types[15,24,53], it is perplexing that mutations in these proteins or RBX1 have not been found in patients with 3M syndrome. And if so, are there essential functions of FBXW8 explaining lack of disease mutations, and by extension does CUL1 also play a role in 3M syndrome? Finally, residue-specific details of how CUL7–RBX1 binds with OBSL1 and CCDC8 and how this contributes to disease are important questions for the future. Our structure of the unusual multidomain CRL7[FBXW8] complex, and finding that this can mediate recruitment of another protein for ubiquitination by neddylated CUL1–RBX1, provide a platform for future studies aimed at understanding these other atypical cullin interactions and mechanisms underlying 3M syndrome (Fig. 5d).

## Online content

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

## Methods

**CRL7^FBXW8 expression constructs.** With the aim of recombinant coexpression of CUL7, RBX1, FBXW8 and SKP1 in mammalian cells we used an approach combining the pEG plasmids from the BacMam system[25] with an adjusted version of the biGBac cloning system[56]. All proteins sequences are of human origin, complementary DNAs were synthesized from Twist Bioscience and variants were produced and verified by standard molecular biology techniques, a list of used primers is in Supplementary Table 1. pEG plasmids for each of the four proteins were cloned, with CUL7 carrying a N-terminal Twin-Strep-tag and FBXW8 a N-terminal GFP-tag, both having a PreScission-cleavage-site between protein of interest and tag. All constructs were full length except RBX1 that was residue 5 to C terminus. Gene-expression cassettes were amplified by PCR with standard forward primers from biGBac system and reverse primers adjusted to be complementary to a sequence downstream of the SV40 terminator. Linearized pBig1 vector and two gene-expression cassettes were fused by a Gibson assembly reaction, resulting in the following circular pBig1 constructs: pBig1a carrying N-terminal twin-Strep-tag CUL7 and RBX1; pBig1a carrying N-terminal twin-Strep-tag CUL7^ΔCPH (residues 389–404 exchanged by a GGGSGGGGSGGGS linker) and RBX1; pBig1b carrying N-terminal GFP-tagged FBXW8 and SKP1. pBig1 constructs were then used to generate bacmids from DH10 EMBacY, which were used to transfect Sf9 cells and amplify baculovirus up to P3. Virus supernatant was sterilized by filtration, then used for subsequent infection for expression.

**CRL7^FBXW8 expression and purification.** Canonical CRL1s are typically assembled from separately purified CUL1–RBX1 and SKP1–F-box protein complexes. Although we were able to express and purify stable CUL7–RBX1 alone, we were not able to obtain a stable and pure SKP1–FBXW8 subcomplex as this readily associated with chaperones. Such chaperone copurification often reflects improper folding that can be overcome by coexpression with other binding proteins. Indeed, we were able to purify homogeneous and stoichiometric CRL7^FBXW8 by coexpressing all four subunits in HEK293S cells in suspension. For expression of CRL7^FBXW8, CRL7^ΔCPH–FBXW8, CUL7–RBX1, 800 ml of HEK cultures were grown to a cellular density of $>3 \times 10^6$ cells per ml and infected with 10% (v/v) P3 of corresponding viruses and incubated at 37 °C. 6 h posttransduction, 10 mM butyrate was added and incubated for 60 h at 30 °C. After that, the collected cells were lysed by sonication in lysis buffer containing 50 mM Tris pH 7.5, 150 mM NaCl, 5 mM DTT, 0.001 mg ml$^{-1}$ Benzonase, EDTA-free protease inhibitor cocktail (one tablet per 50 ml of buffer). Insoluble material was removed by centrifugation at 50,000g and supernatant was incubated with Strep-Tactin resin in-batch rocking for 15 min at 4 °C. The resin was then transferred to a column and washed with 25 CV of Wash Buffer containing 50 mM Tris pH 7.5, 150 mM NaCl, 5 mM DTT. Protein was eluted with 10 CV of Elution Buffer containing 2.5 mM D-desthiobiotin in the wash buffer and sequentially cleaved with 3C protease overnight at 4 °C, followed by Anion exchange chromatography. Protein containing fractions were pooled and concentrated to <500 μl using a 100 kDa cutoff concentrator (Amicon) and final polishing was performed by SEC on a Superose 6 Increase 10/300 GL column (GE Healthcare) with Wash Buffer. Peak fractions containing CRL7^FBXW8 were pooled, concentrated to around 1.7 mg ml$^{-1}$ and used immediately for cryo-EM grid preparation.

**Cloning, expression and purification of TP53.** His-lipoyl domain-tagged TP53 was cloned into a pRSF vector, expressed in *Escherichia coli* BL21 Rosetta (DE3) at 20 °C overnight. Cells were lysed in 50 mM Tris pH 8.0, 300 mM NaCl, 0.01% NP-40, EDTA-free protease inhibitor cocktail (one tablet per 50 ml of buffer), 5 mM β-mercaptoethanol. After sonication, the lysate was centrifuged for 20 min at 15,000g and the supernatant was loaded onto a Ni-NTA column. His-tagged TP53 was eluted with increasing imidazole gradient and dialyzed overnight in wash buffer (50 mM Tris pH 7.5, 150 mM NaCl, 5 mM DTT) in the presence of tobacco etch virus (TEV) protease. Cleaved TP53 ran as a tetramer on a Superose 6 Increase 10/300 GL column (GE Healthcare) and was separated from the His-tagged lipoyl domain.

**Generation of TAMRA-TP53 fusion via Sortase-mediated transpeptidation reaction.** TAMRA-labeled TP53 was generated in a Sortase-catalyzed reaction by fusing a TAMRA–LPETGG peptide (obtained from Max Planck Institute of Biochemistry Core Facility) with the N terminus of TP53, which retains a glycine after TEV cleavage. Reaction was incubated with concentrations of 50 μM TP53, 300 μM TAMRA–LPETGG peptide and 5 μM SortaseA-6xHis harboring activity enhancing mutations[57] for 2 h in 50 mM Tris pH 8.0 150 mM NaCl 10 mM CaCl$_2$, removed SortaseA by retention on nickel resin and further purified by SEC in 50 mM Tris pH 7.5, 150 mM NaCl, 1 mM DTT.

**Cloning, expression and purification of ubiquitination and neddylation machinery and variants.** All proteins are of human origin and were cloned and verified by standard molecular biology techniques. Full-length CUL1, CUL1^K720R, glutathione *S*-transferase (GST)-TEV site (TEV)-RBX1 (residue 5 to C terminus), CUL5, GST-TEV-RBX2 (residue 5 to C terminus) and GST-TEV-UBA1, were expressed in *Trichoplusia ni* High-Five insect cells, with CUL and corresponding RBX protein, being coexpressed[35,41,52]. CUL1 N-terminal region (residues 1 to 410

encompassing CUL1's CR1, CR2 and CR3 domains), UBE2A, UBE2B, UBE2C, UBE2D1, UBE2D2, UBE2D3, UBE2D4, UBE2E1, UBE2E2, UBE2E3, UBE2G1, UBE2G2, UBE2H, UBE2I, UBE2J1, UBE2J2, UBE2K, UBE2L3, UBE2N, UBE2Q2, UBE2R1, UBE2R2, UBE2S, UBE2T, UBE2V1, UBE2V2 and the neddylation components UBE2M and NAE1-UBA3 were expressed in *E. coli* Rosetta (DE3) cells as GST-thrombin or TEV fusion proteins. CUL1^332-C (from residue 332 to the C terminus) was coexpressed with GST-TEV-RBX1 in *E. coli* Rosetta (DE3) cells.

Fusion proteins were subjected to GST-affinity chromatography, eluted by overnight protease cleavage on beads and further purified by ion-exchange and SEC in 25 mM HEPES, 150 mM NaCl, 1 mM DTT[36,40].

For fluorescent labeling GST-TEV fusion proteins of NEDD8 and ubiquitin (wild-type and K48R) carried an additional cysteine in the N-terminal region were purified similarly[40]. Purified NEDD8 and ubiquitin variants were then incubated 30 min on ice with addition of 10 mM DTT to ensure full reduction for single cysteine labeling. Proteins were buffer exchanged into 25 mM HEPES pH 7.5, 150 mM NaCl using NAP-5 columns, 1 mM Fluorecein-5-maleimide was added to 200 μM protein and incubated for 2 h at room temperature. To remove free unreacted dye the reaction mix was buffer exchanged a second time using a NAP-5 column in 25 mM HEPES 7.5, 150 mM NaCl, 1 mM DTT and fluorescently tagged protein was purified by SEC in the same buffer[40].

Neddylation of CUL1–RBX1 was carried out in 25 mM HEPES pH 7.5, 150 mM NaCl, 2.5 mM MgCl, 1.5 mM ATP, with 0.04× NAE1-UBA3 and UBE2M, 2.5× NEDD8 (wild-type or Q40E mutant) for 5 h at 4 °C. NEDD8–CUL1–RBX1 was purified by cation exchange and SEC in 25 mM HEPES pH 7.5, 150 mM NaCl, 1 mM DTT. Neddylation of CUL5–RBX2 was performed similarly using UBE2F instead of UBE2M (ref. [40]).

His-MBP-TEV β-TRCP1 (monomeric form, from residue 175 to C terminus) was expressed and purified in complex with 'SKP1^ΔΔ' (SKP1 harboring two internal deletions, of residues 38–43 and 71–82). The complex was purified by nickel affinity chromatography, followed by TEV cleavage, ion-exchange and SEC in 25 mM HEPES 7.5, 150 mM NaCl, 1 mM DTT[35]. This complex is referred to as SKP-β-TRCP1 in the main text and figures.

**Cryo-EM sample preparation and data collection for CRL7^FBXW8.** R1.2/1.3, 200 mesh holey carbon grids (Quantifoil) were glow discharged, 4 μl of freshly purified CRL7^FBXW8 was applied at 100% humidity and 4 °C in a Vitrobot Mark IV (Thermo) and plunge-frozen into liquid ethane (5 s incubation time, blot force 3–4, blot time 4 s). High-resolution cryo-EM data were collected on a Titan Krios electron microscope operating at 300 kV, equipped with a post-GIF Gatan K3 Summit direct electron detector (counting mode) using SerialEM[58]. Videos were recorded at a nominal magnification of ×105,000 (0.8512 Å per pixel at the specimen level), with target defocus ranging between −0.7 and −2.8 μm and total exposure of roughly 60 e$^-$/Å$^2$ fractionated over 40 frames.

**Cryo-EM data processing.** RELION 3.0 (ref. [59]) was used for motion-correction and dose weighting, the contrast transfer function was estimated using CTFFIND-4.1 (ref. [60]) and particles were picked using Gautomatch (K. Zhang, MRC Laboratory of Molecular Biology) with a template based on previous screening datasets. 19,499 micrographs with a maximum resolution estimate better than 5 Å were imported into RELION v.3.1 (ref. [59]), from which roughly 6.05 million particles were extracted applying sixfold binning. These were subjected to several rounds of 2D classification, followed by initial model generation and 3D classification. Then 355,547 particles from the best 3D class were reextracted at full pixel size, followed by masked 3D autorefinement, producing a reconstruction at 2.82 Å overall resolution. RELION[59] postprocessing and DeepEMhancer[54] were used for sharpening of the final map.

**Model building and refinement.** The CRL7^FBXW8 structure was traced de novo using Coot (v.0.8.9.2)[61] and Phenix.refine was used for real space refinement[62]. Residues were placed starting with the best resolved cullin repeats and WD40 domain of FBXW8, and sequentially filled in the rest of the density by seeking landmark residues (tryptophan, phenylalanine, tyrosine, proline) in the sequences with corresponding well-resolved densities. These were used in addition to secondary structure prediction to guide sequence assignment, loop building and helical packing. RBX1 was built based on several of the published structures in the Research Collaboratory for Structural Bioinformatics (RCSB) (for example, PDB 4P5O)[40]. The density representing SKP1 allowed placement of helices but was not resolved well enough for unambiguous placement of side chains except in the interaction surface between the F-box domain of FBXW8 (Extended Data Fig. 3a). A SKP1 crystal structure was fitted in the density and kept unchanged except deleting loops that had no density in the cryo-EM map. The resolution of cullin repeats and WD40 domain of FBXW8 was better than 3 Å in most parts allowing for side-chain placement (Extended Data Fig. 3d,e). However, there was an additional poorly resolved domain near the N terminus of CUL7 domain, which was not assigned to any part of CRL7^FBXW8 (Extended Data Fig. 3c). The same is true for a patch of density adjacent to the RBX1 RING domain that could resemble a helix (Extended Data Fig. 3b). After the structure was determined, AlphaFold was released and enabled comparison of the AlphaFold-derived model to the experimentally derived structure[34].

**Cryo-EM sample preparation and data collection for CRL7[FBXW8]-CUL1 N-terminal region.** The CRL7[FBXW8]–CUL1 N-terminal region complex was prepared by mixing the purified proteins 15 min before plunging and incubating them on ice. Cryo-EM data were collected on a Glacios electron microscope operating at 200 kV, equipped Gatan K2 (counting mode) using SerialEM[58]. Videos were recorded at a nominal magnification of ×22,000 (1.885 Å per pixel at the specimen level), with target defocus ranging between −1.2 and −3.3 μm and total exposure of roughly $60 e^-/Å^2$ fractionated over 40 frames. Plunging and data processing were performed similarly to CRL7[FBXW8]. Maps and structures in the paper were rendered by Pymol v.2.3.4, Chimera v.1.13.1 or ChimeraX v.1.2.5.

**Neddylation and ubiquitination assays.** Neddylation assays were performed by mixing 0.5 μM NAE1-UBA3, 1.2 μM UBE2M or UBE2F, 1.5 μM E3 and 20 μM GST-NEDD8 or 10 μM Fluorescein-labeled NEDD8 in a buffer of 25 mM HEPES pH 7.5, 150 mM NaCl, 2.5 mM $MgCl_2$ and 1.5 mM ATP. Reactions were quenched by the addition of SDS–PAGE sample buffer at time points indicated in the corresponding figures. SDS–PAGE gel was scanned on an Amersham Typhoon Imager (Cy2 channel) to visualize fluorescent NEDD8. Autoubiquitination assays were performed by mixing 0.2 μM UBA1, 1 μM E2, 10 μM Fluorescein-labeled ubiquitin and 0.5 μM E3 in a buffer of 25 mM HEPES pH 7.5, 150 mM NaCl, 2.5 mM $MgCl_2$, 1.5 mM ATP. Screening for various E2s was performed under the same conditions. Reactions were quenched after 10 min (or at time points indicated in the corresponding figures) by addition of SDS–PAGE sample buffer. SDS–PAGE gel was scanned on an Amersham Typhoon Imager (Cy2 channel) to visualize fluorescent ubiquitin.

Ubiquitination of TP53 was monitored using TAMRA-labeled TP53. 0.5 μM TAMRA-TP53, 0.5 μM E3, 1 μM UBE2D3, 0.2 μM UBA1 and 20 μM ubiquitin were mixed in a buffer of in 25 mM HEPES pH 7.5, 150 mM NaCl, 2.5 mM $MgCl_2$ and 1.5 mM ATP at room temperature for 10 min. Reactions were quenched after 10 min by addition of SDS–PAGE sample buffer. SDS–PAGE gel was scanned on an Amersham Typhoon Imager (Cy3 channel) to visualize fluorescent TAMRA-TP53.

Di-ubiquitin synthesis was assayed in a pulse-chase format. Pulse of a E2 loaded ubiquitin (E2~UB) was generated by incubating 10 μM E2 with 10 μM fluorescein-labeled K48R ubiquitin in the presence of 0.2 μM UBA1 in a buffer containing 50 mM HEPES, pH 7.5, 50 mM NaCl, 5 mM $MgCl_2$ and 5 mM ATP. The reaction was incubated for 30 min at room temperature, and was quenched with 25 mM EDTA for 5 min on ice. The quenched pulse was diluted and added to chase reactions containing a mixture of E3, and wild-type ubiquitin to start the reaction so that final concentrations of the reaction would be 0.5 μM E2~UB, 0.5 μM E3 and 40 μM wild-type ubiquitin in a buffer containing 25 mM HEPES, pH 7.5, 50 mM NaCl. The reactions were quenched with nonreducing SDS–PAGE sample buffer after incubation at room temperature for the indicated times points. SDS–PAGE gel was scanned on an Amersham Typhoon Imager (Cy2 channel) to visualize fluorescent ubiquitin.

**SEC comigration experiments.** To examine comigration during SEC, proteins were mixed at equimolar concentrations of 1.5 μM for at least 10 min on ice. Then 50 μl of each mix was loaded onto a Superose 6, 5/150GL column (GE), with a running buffer of 50 mM Tris pH 7.5, 150 mM NaCl and 1 mM DTT. Fractions from SEC were analyzed by SDS–PAGE, with proteins detected by Coomassie staining. Chromatograms were processed in GraphPad Prism v.9.2.0.

**Reporting summary.** Further information on research design is available in the Nature Research Reporting Summary linked to this article.

## Data availability

The atomic coordinates and cryo-EM maps have been deposited in the RCSB with accession codes PDB ID 7Z8B and EMDB with codes EMD-14547 (CRL7[FBXW8]) and EMD-14558 (CRL7[FBXW8]–CUL1[NTD]). Datasets from PDB used in this study include: 2OVP, 1NEX, 4P5O, 6O60, 6TTU, 1LDJ, 2JNG and 7ONI. Raw gels are provided as source data. Source data are provided with this paper.

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

## Acknowledgements

We thank J. Frye for initial attempts to produce CUL7–RBX1, B. Bräuning for assistance in establishing expression systems for CUL7–RBX1 and CRL7[FBXW8] and with cryo-EM, J. Rajan Prabu for assistance with cryo-EM data processing and model validation, and J. Kellermann, B. Bräuning, D. Sherpa, J. Botsch, D. Horn-Ghetko, S. Kostrhon, K. Swatek and J. Liwocha for assistance, helpful discussions and/or critical reading of the manuscript. We thank D. Bollschweiler and T. Schäfer of the cryo-EM facility at Max Planck Institute of Biochemistry. This study was supported by the Max Planck Gesellschaft and a H2020 grant from the European Research Council (ERC grant no. 789016-NEDD8Activate, B.A.S.) and National Institutes of Health grant no. CA068377 (Y.X., now to W. Marzluff).

## Author contributions

L.V.M.H., Y.X. and B.A.S. conceived the project. L.V.M.H., K.B., S.v.G. and M.K. generated protein complexes. L.V.M.H. performed enzyme assays and generated cryo-EM samples. L.V.M.H. collected, processed and refined cryo-EM data. L.V.M.H. built and refined the structure. B.A.S. and L.V.M.H. analyzed the data and prepared the manuscript with input from the other authors. B.A.S. supervised the project.

## Funding

## Competing interests

B.A.S. is a member of the scientific advisory boards of Interline Therapeutics and BioTheryx, and is a coinventor of intellectual property related to DCN1 inhibitors licensed to Cinsano. Y.X. is Chief Scientific Officer of Cullgen. The remaining authors declare no competing interests.

## Additional information

**Extended data** are available for this paper at https://doi.org/10.1038/s41594-022-00815-6.

**Correspondence and requests for materials** should be addressed to Brenda A. Schulman.

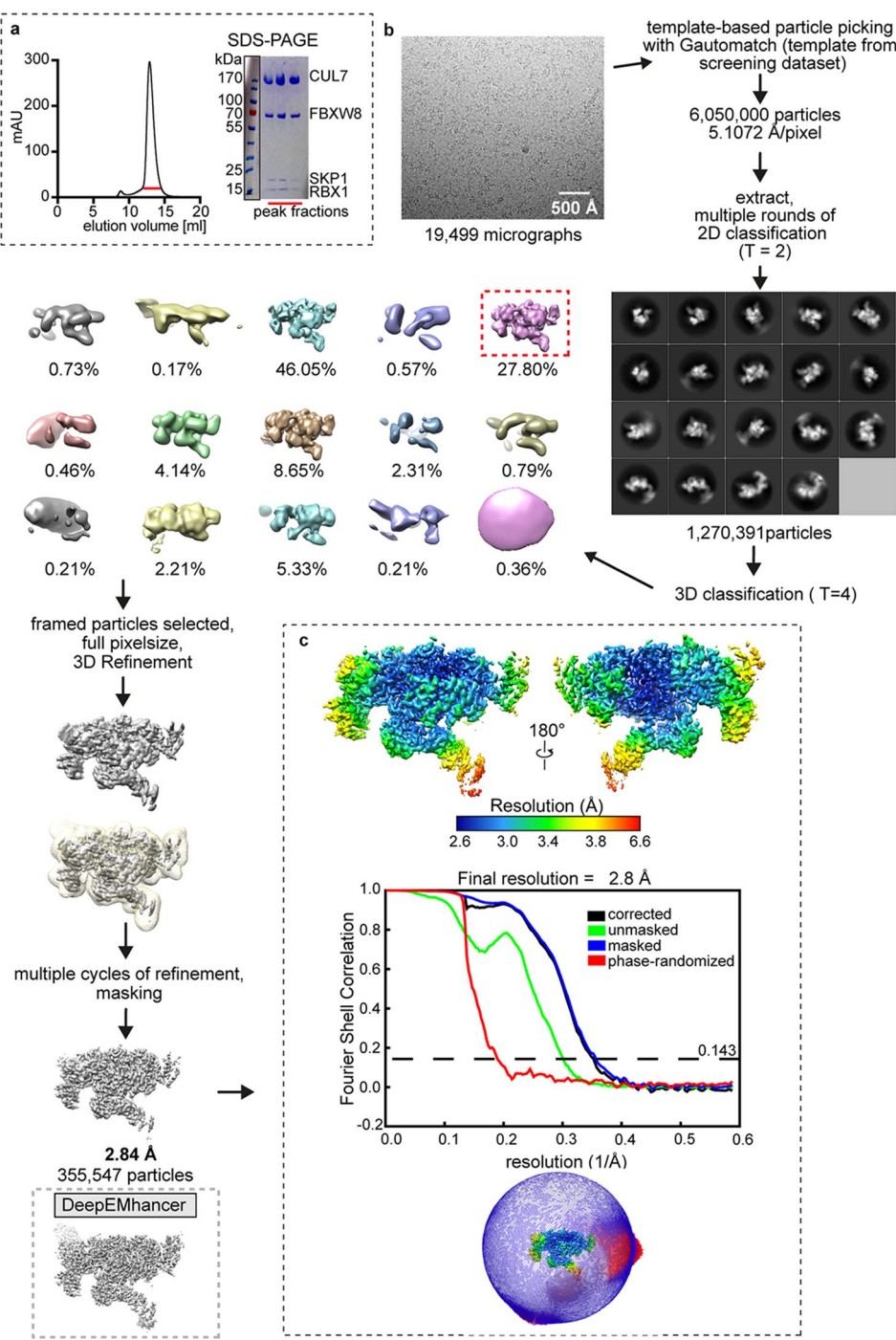

**Extended Data Fig. 1 | Sample preparation and Cryo-EM processing flowchart for CRL7^FBXW8. a**, Final size-exclusion chromatography profile from purification of CRL7^FBXW8, with the corresponding peak fractions visualized on a Coomassie-stained SDS-PAGE gel. **b**, Cryo-EM image processing flowchart for CRL7^FBXW8. **c**, Local resolution map in two views, Fourier Shell Correlation (FSC) curve with an overall resolution at 2.8 Å at the gold standard FSC = 0.143 and the angular distribution of the final reconstruction.

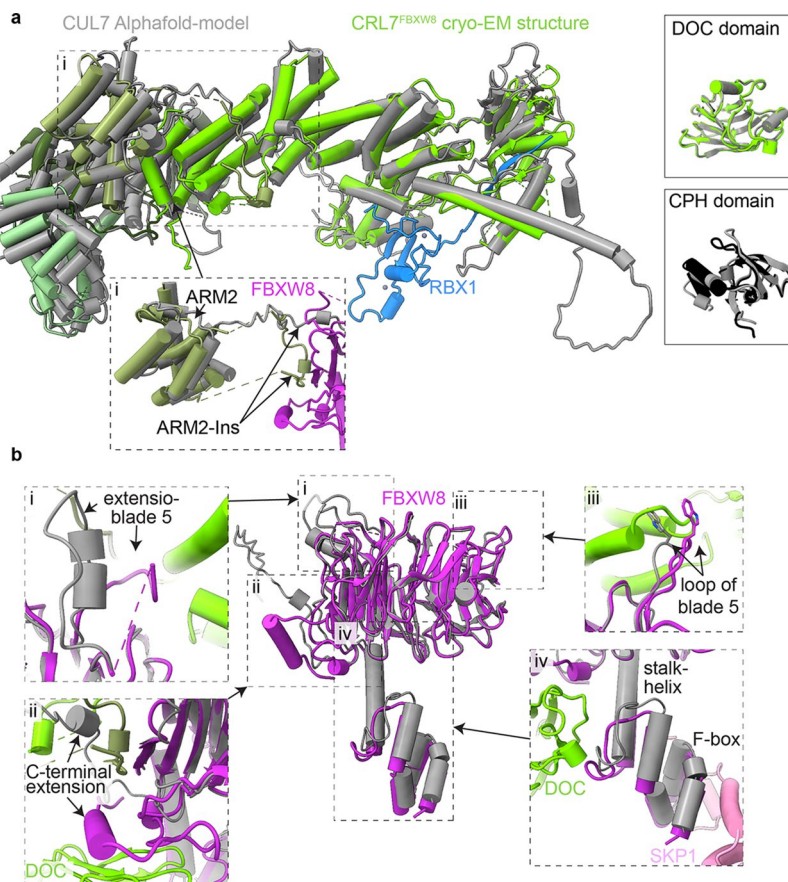

**Extended Data Fig. 2 | Comparison of CRL7$^{FBXW8}$ cryo-EM structure with predicted models from AlphaFold. a,** Overlay of CUL7-RBX1 from the CRL7$^{FBXW8}$ structure with the AlphaFold-generated model of CUL7. For clarity, FBXW8-SKP1 was excluded in this overlay. DOC and CPH domain had no interfaces to the rest of CUL7 but are loosely-tethered domains in the AlphaFold-model, and hence are overlaid separately. The CPH domain is not visible in the CRL7$^{FBXW8}$ cryo-EM structure. Thus, the published NMR structure (PDB:2JNG) was overlaid with the AlphaFold model[30,34]. **b,** Overlay of FBXW8-SKP1 from CRL7$^{FBXW8}$ with AlphaFold model of FBXW8. (i-iv) Close-ups of individual regions with CUL7 and SKP1 structure added to visualize structural differences at binding interfaces: (i) FBXW8 blade 5 extension; (ii) FBXW8 C-terminal extension; (iii) FBXW8 loop of blade 5; (iv) FBXW8's stalk-helix and F-box.

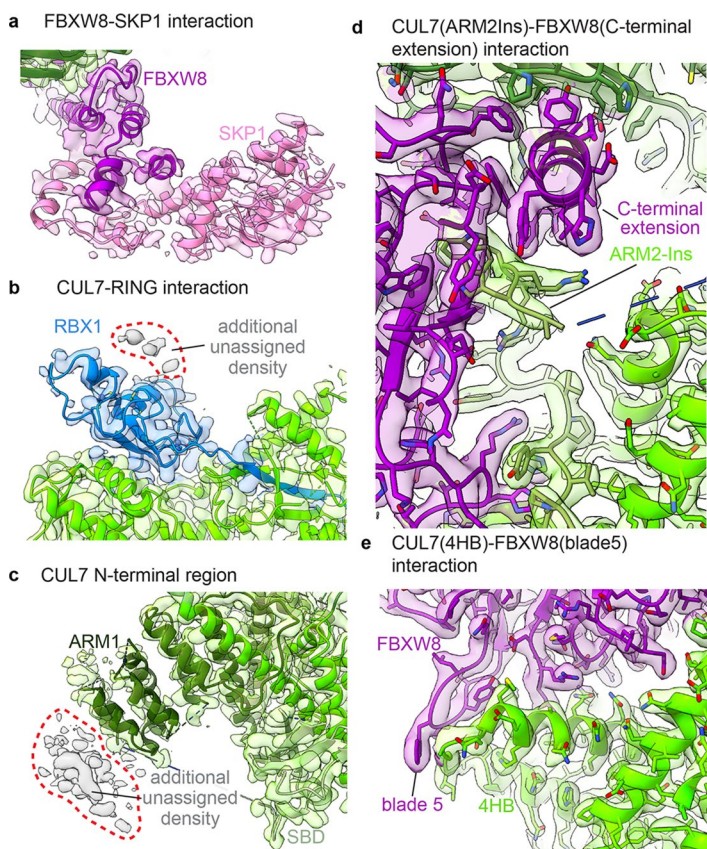

**Extended Data Fig. 3 | Cryo-EM map quality analysis. a**, Cryo-EM density and structure in the region of FBXW8-SKP1 interface. The crystal structure of SKP1 (PDB:6O60) was fitted in the weak corresponding density. **b**, Cryo-EM density and structure in the region of RBX1 with additional unassigned density in grey adjacent to RBX1 RING domain. **c**, Cryo-EM density with structure in the N-terminal region of CUL7 with some unassigned density next to the ARM1 domain. **d**, Well-resolved interface of FBXW8 C-terminal extension and CUL7s ARM2-Ins. **e**, Well-resolved interface of FBXW8s blade 5 loop with CUL7s 4HB domain.

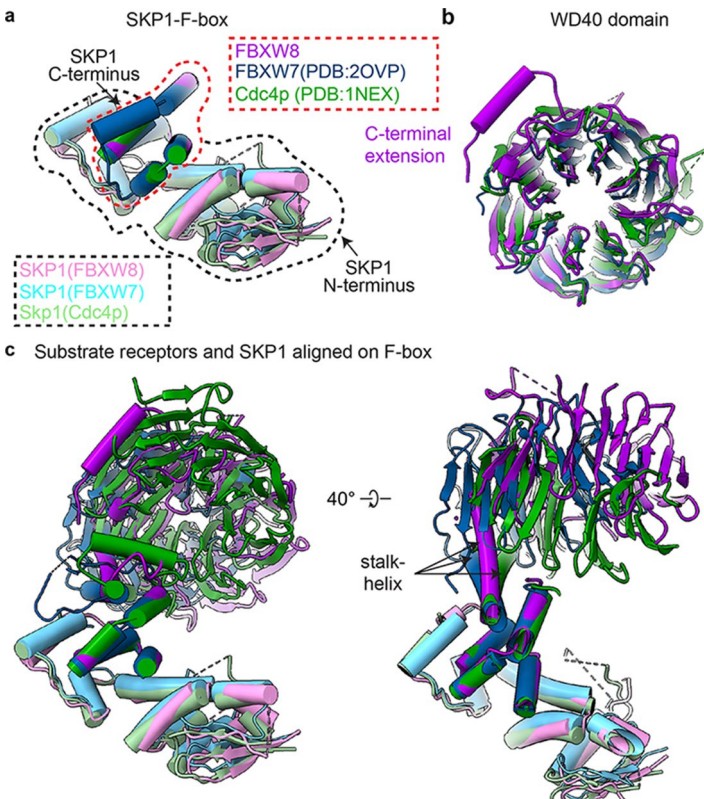

**Extended Data Fig. 4 | FBXW8 comparison with other WD40 F-box proteins. a**, F-box overlay of FBXW8-SKP1, FBXW7-SKP1 (PDB:2OVP) and Cdc4p-Skp1 (PDB:1NEX)[37,38]. **b**, WD40 domain overlay of FBXW8, FBXW7 (PDB:2OVP) and Cdc4p (PDB:1NEX)[37]. **c**, Overlay of F-box protein-SKP1 complexes aligned on F-box.

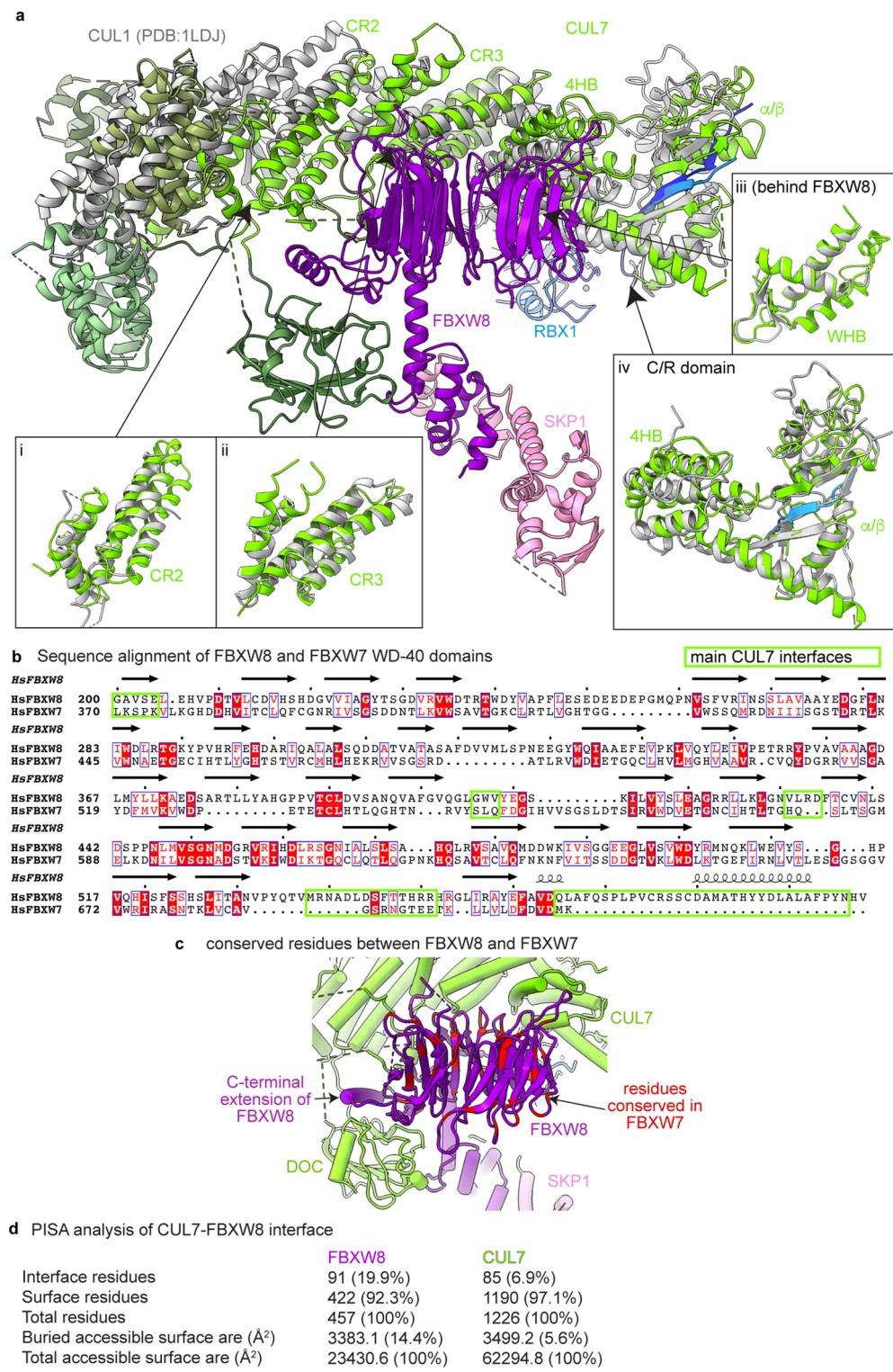

**Extended Data Fig. 5 | CRL7^FBXW8 compared to various CRL1 complexes. a**, Overlay of CRL7^FBXW8 with CUL1–RBX1 (PDB:1LDJ)[6]. (i) Overlay of CR2 domains. (ii) Overlay of CR3 domains. (iii) Overlay of WHB domains. (iv) Overlay of 4HB and α/β domains. **b**, Sequence alignment of WD-40 domains from human FBXW8 and FBXW7. Identical residues are highlighted with red boxes, similar residues with blue boxes and red font. Secondary structure assignments for FBXW8 are shown above the sequences. Green boxes indicate the predominant FBXW8 regions forming the interfaces with CUL7. Structure-based alignments were calculated with Chimera and visualized with Espript (https://espript.ibcp.fr/ESPript/ESPript/). **c**, Residues that are identical in FBXW8 and FBXW7 are plotted in red on the structure of FBXW8 in the CRL7^FBXW8 complex. **d**, Analysis of FBXW8-CUL7 interface by QtPISA v2.1.0, showing interface residues and buried surface area.

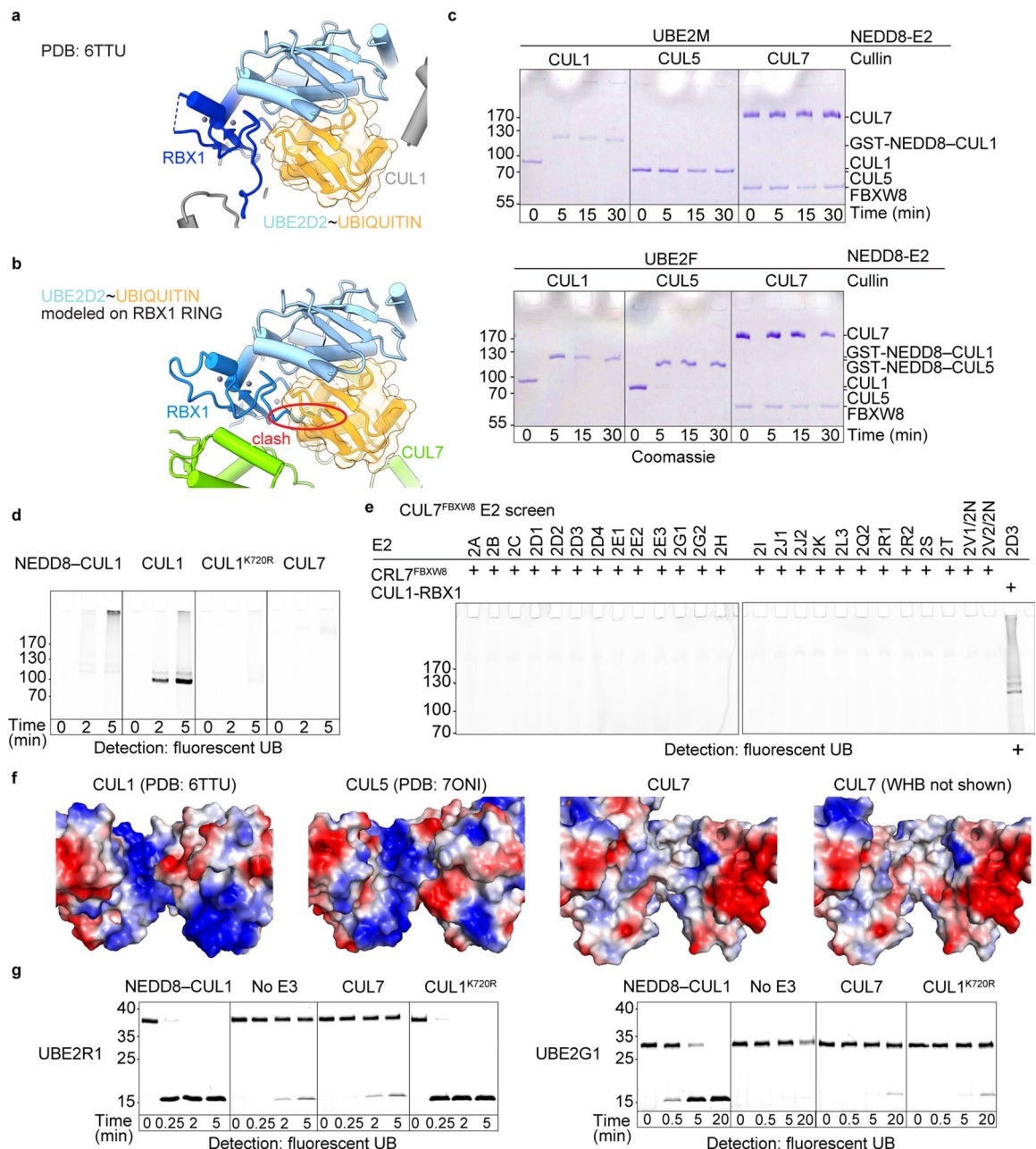

**Extended Data Fig. 6 | CUL7-RBX1 is not autoubiquitinated under conditions allowing efficient autoubiquitination of CUL1. a**, Closeup view of CUL1-RBX1-E2 (UBE2D2)~UB (PDB:6TTU)[35]. **b**, UBE2D2-UB (of CRL1[β-TRCP], PDB:6TTU) clashes with CRL7[FBXW8] when RBX1 RING domains from the two complexes are aligned. **c**, In vitro neddylation assay using GST-NEDD8, CRL7[FBXW8] or the indicated cullin-RING complexes, a NEDD8 E2 (UBE2M or UBE2F), and NEDD8 E1 (NAE-UBA3), visualized by size shift in Coomassie-stained SDS-PAGE-gel (samples of both gels derived from the same experiment and gels were processed in parallel, n = 2 technically independent experiments). **d**, Autoubiquitination assay detecting fluorescently-labelled ubiquitin of the indicated E3 complexes, performed with UBE2D3 as E2 and UBA1 as E1 (n = 2 technically independent experiments). **e**, Assay for CRL7[FBXW8] autoubiquitination with the indicated E2s, UBA1 as E1, and fluorescently labelled ubiquitin, except the last positive control lane (labeled + below) where CUL1-RBX1 was used instead of CRL7[FBXW8] (samples of both gels derived from the same experiment and gels were processed in parallel, n = 2 technically independent experiments). **f**, Electrostatic potential of the surfaces corresponding to the 'basic canyon' from CUL1 (PDB: 6TTU), CUL5 (PDB: 7ONI) and CUL7 (this study), where blue is positively charged and red is negatively charged. The electrostatic potential was calculated in PYMOL v2.3.4. Additionally, the same plot is shown for CUL7 without the WHB-domain and its linker region, which partly cover the region corresponding to the 'basic canyon' involved in UBE2R1 binding to other cullins[45]. **g**, Pulse-chase assays for di-ubiquitin synthesis, monitoring transfer of fluorescent ubiquitin from the active site of either UBE2R1 (left panel) or UBE2G1 (right panel) to free unlabeled ubiquitin without E3 or with indicated RBX1-bound cullin complexes (samples of all gels derived from the same experiment and gels were processed in parallel, n = 2 technically independent experiments).

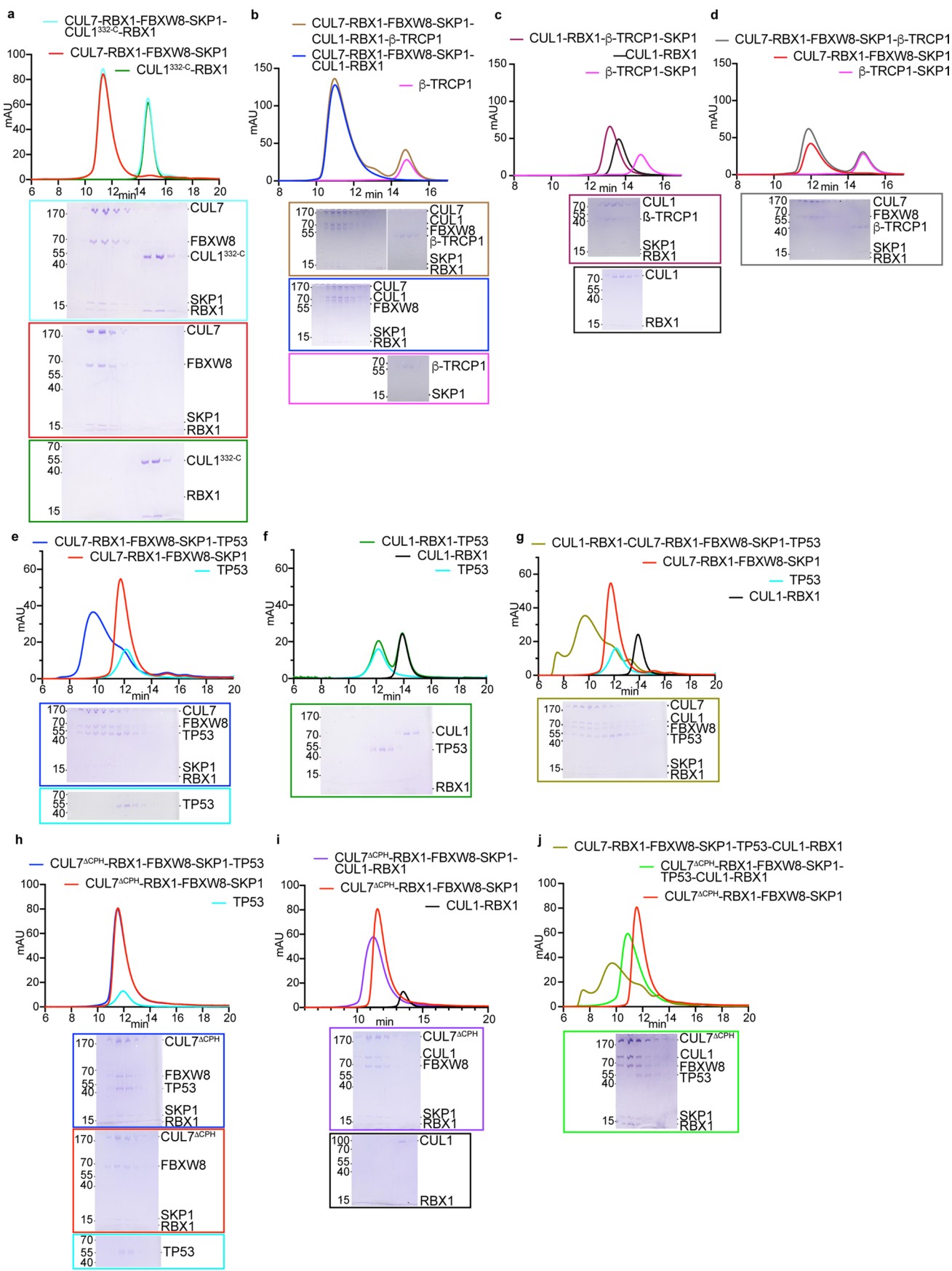

**Extended Data Fig. 7 | See next page for caption.**

**Extended Data Fig. 7 | Size-exclusion chromatography co-migration of CRL7$^{FBXW8}$ and its binding partners.** Experiments assay protein migration on a Superose 6, 5/150GL column. Size exclusion chromatography (SEC) profiles are shown above Coomassie-stained SDS-PAGE gel analyses of corresponding elution fractions. For size and clarity reasons the experiments are split in multiple panels. Some SEC profiles are used in multiple panels for comparison. To avoid showing the corresponding gels multiple times they are only shown at the first use (compare also Fig. 5a) **a**, Samples were purified CRL7$^{FBXW8}$ or CUL1$^{332-C}$-RBX1 (CUL1 residues 332 through the C-terminus bound to RBX1) alone or mixed before subjecting to SEC. **b**, Samples were purified CRL7$^{FBXW8}$-CUL1-RBX1 and SKP1-β-TRCP1 alone or mixed before subjecting to SEC. **c**, Samples were purified CUL1-RBX1 and SKP1-β-TRCP1 alone or mixed before subjecting to SEC. **d**, Samples were purified CRL7$^{FBXW8}$ and SKP1-β-TRCP1 alone or mixed before subjecting to SEC. **e**, Samples were purified CRL7$^{FBXW8}$ and TP53 alone or mixed before subjecting to SEC. **f**, Samples were purified CUL1-RBX1 and TP53 alone or mixed before subjecting to SEC. **g**, Samples were purified CRL7$^{FBXW8}$, CUL1-RBX1, and TP53 alone or the indicated mixture generated before subjecting to SEC. **h**, Samples were purified CRL7$^{\Delta CPH-FBXW8}$ (a mutant complex with residues 389–404 in CUL7's CPH domain exchanged by GS-linker sequence) and TP53 alone or mixed before subjecting to SEC. **i**, Samples were purified CRL7$^{\Delta CPH-FBXW8}$ and CUL1-RBX1 alone or mixed before subjecting to SEC. **j**, Samples were CRL7$^{\Delta CPH-FBXW8}$, CRL7$^{\Delta CPH-FBXW8}$-CUL1-RBX1-TP53, and CRL7$^{FBXW8}$-CUL1-RBX1-TP53.

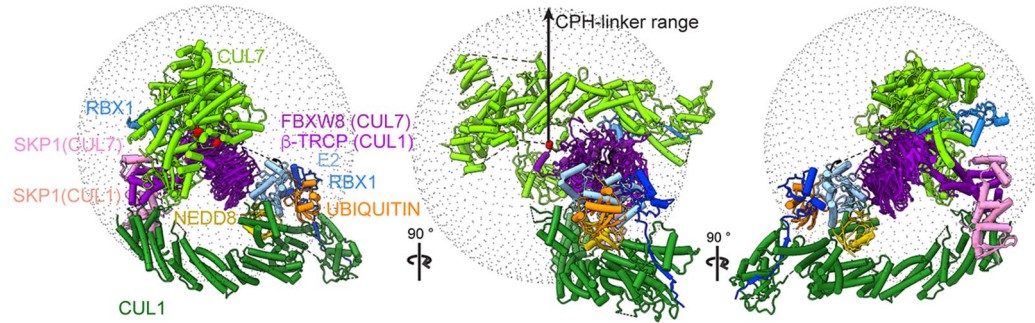

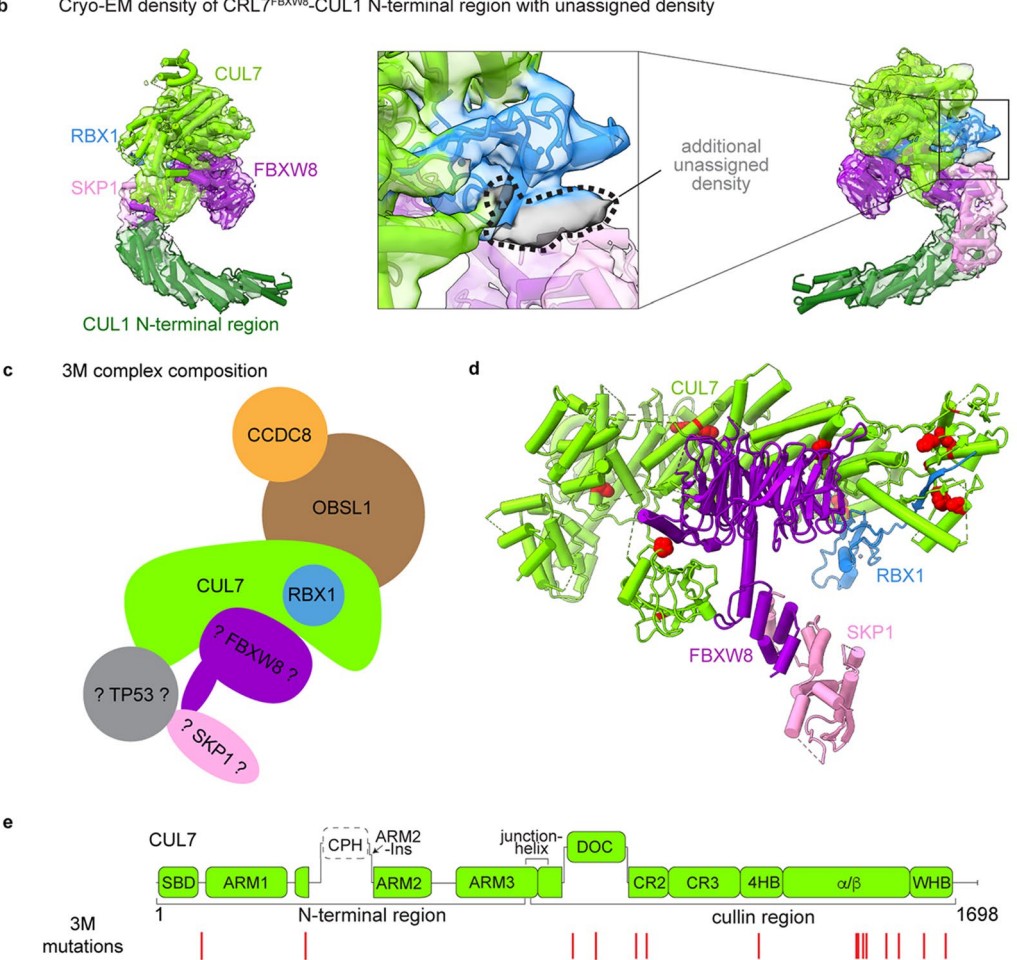

**Extended Data Fig. 8 | Overlay of structure showing substrate ubiquitination by CRL1^β-TRCP (PDB:6TTU) on SKP1 from CRL7^FBXW8 and structural map of CUL7 3 M disease mutations. a**, Three different views of CRL1^β-TRCP (PDB:6TTU)[35] superimposed on SKP1 of CRL7^FBXW8. Spheres indicate possible range (roughly 80 Å) of the flexible linker to the CPH domain, which is the reported TP53 binding platform. Red dots indicate the last visible residues preceding and following the CPH-domain (D314 and A461). The sphere is centered on A461, which marks the origin of the shorter linker. **b**, Structure of CRL7^FBXW8 and CUL1 N-terminal region (residues 1-410) docked in corresponding cryo-EM map in two views matching those in **a**. Close-up visualizes additional unassigned density adjacent to RBX1 RING domain, which is more pronounced than in the CRL7^FBXW8 map. This density potentially corresponds to an unassigned portion of CUL7 C-terminal domain. **c**, Schematic representation of 3 M complex. 3 M disease is caused by mutations in OBSL1, CCDC8 and CUL7 which form the 3 M complex. FBXW8-SKP1 and TP53 may bind and/or be parts of functional 3 M complex assemblies. **d**, 3 M disease mutations indicated in red spheres shown on the structure of CRL7^FBXW8. Note that Q93H and S1536L could not be visualized because the corresponding cryo-EM density was too weak to build these residues. **e**, 3 M disease mutations indicated in red on the domain map of CUL7.

# Reporting Summary

## Statistics

For all statistical analyses, confirm that the following items are present in the figure legend, table legend, main text, or Methods section.

| n/a | Confirmed | |
|---|---|---|
| ☐ | ☒ | The exact sample size (*n*) for each experimental group/condition, given as a discrete number and unit of measurement |
| ☐ | ☒ | A statement on whether measurements were taken from distinct samples or whether the same sample was measured repeatedly |
| ☒ | ☐ | The statistical test(s) used AND whether they are one- or two-sided<br>*Only common tests should be described solely by name; describe more complex techniques in the Methods section.* |
| ☒ | ☐ | A description of all covariates tested |
| ☒ | ☐ | A description of any assumptions or corrections, such as tests of normality and adjustment for multiple comparisons |
| ☒ | ☐ | A full description of the statistical parameters including central tendency (e.g. means) or other basic estimates (e.g. regression coefficient) AND variation (e.g. standard deviation) or associated estimates of uncertainty (e.g. confidence intervals) |
| ☒ | ☐ | For null hypothesis testing, the test statistic (e.g. *F*, *t*, *r*) with confidence intervals, effect sizes, degrees of freedom and *P* value noted<br>*Give P values as exact values whenever suitable.* |
| ☒ | ☐ | For Bayesian analysis, information on the choice of priors and Markov chain Monte Carlo settings |
| ☒ | ☐ | For hierarchical and complex designs, identification of the appropriate level for tests and full reporting of outcomes |
| ☒ | ☐ | Estimates of effect sizes (e.g. Cohen's *d*, Pearson's *r*), indicating how they were calculated |

*Our web collection on statistics for biologists contains articles on many of the points above.*

## Software and code

Policy information about availability of computer code

| Data collection | Gel imaging: Amersham Imager 600, Amersham Typhoon;  Cryo-EM: SerialEM v3.8.0-b5, FEI EPU v2.7.0 |
|---|---|
| Data analysis | Assay Analysis: GraphPad Prism v9.2.0;  Cryo-EM: RELION v3.0, RELION v3.1, Gautomatch v0.56, CTFFIND v4.1;  Structure Analysis and Visualization: Chimera v1.13.1, ChimeraX v1.2.5, PyMol v2.3.4; QtPISA v2.1.0; Model Building: COOT v0.8.9.1, Phenix.refine v1.17.1, DeepEMhancer (https://github.com/rsanchezgarc/deepEMhancer) |

For manuscripts utilizing custom algorithms or software that are central to the research but not yet described in published literature, software must be made available to editors and reviewers. We strongly encourage code deposition in a community repository (e.g. GitHub). See the Nature Portfolio guidelines for submitting code & software for further information.

## Data

Policy information about availability of data

All manuscripts must include a data availability statement. This statement should provide the following information, where applicable:
- Accession codes, unique identifiers, or web links for publicly available datasets
- A description of any restrictions on data availability
- For clinical datasets or third party data, please ensure that the statement adheres to our policy

The atomic coordinates and cryo-EM maps have been deposited in the RCSB with accession codes PDB ID 7Z8B and EMDB with codes EMD-14547 (CRL7FBXW8) and EMD-14558 (CRL7FBXW8-CUL1NTD).  Datasets from PDB used in this study include: 2OVP, 1NEX, 4P5O, 6O60, 6TTU, 1LDJ, 2JNG and 7ONI.  Raw gels are provided as source data.

# Field-specific reporting

Please select the one below that is the best fit for your research. If you are not sure, read the appropriate sections before making your selection.

☒ Life sciences ☐ Behavioural & social sciences ☐ Ecological, evolutionary & environmental sciences

For a reference copy of the document with all sections, see [nature.com/documents/nr-reporting-summary-flat.pdf](nature.com/documents/nr-reporting-summary-flat.pdf)

# Life sciences study design

All studies must disclose on these points even when the disclosure is negative.

| | |
|---|---|
| Sample size | Sample size calculations were not performed. Based on previous experience, at least two independent replicates were carried out for all functional assays. |
| Data exclusions | No data were excluded. |
| Replication | All experiments were performed at least twice, with numerous controls. All attempts at replication were successful. |
| Randomization | No grouped samples. |
| Blinding | No grouped samples. |

# Behavioural & social sciences study design

All studies must disclose on these points even when the disclosure is negative.

| | |
|---|---|
| Study description | *Briefly describe the study type including whether data are quantitative, qualitative, or mixed-methods (e.g. qualitative cross-sectional, quantitative experimental, mixed-methods case study).* |
| Research sample | *State the research sample (e.g. Harvard university undergraduates, villagers in rural India) and provide relevant demographic information (e.g. age, sex) and indicate whether the sample is representative. Provide a rationale for the study sample chosen. For studies involving existing datasets, please describe the dataset and source.* |
| Sampling strategy | *Describe the sampling procedure (e.g. random, snowball, stratified, convenience). Describe the statistical methods that were used to predetermine sample size OR if no sample-size calculation was performed, describe how sample sizes were chosen and provide a rationale for why these sample sizes are sufficient. For qualitative data, please indicate whether data saturation was considered, and what criteria were used to decide that no further sampling was needed.* |
| Data collection | *Provide details about the data collection procedure, including the instruments or devices used to record the data (e.g. pen and paper, computer, eye tracker, video or audio equipment) whether anyone was present besides the participant(s) and the researcher, and whether the researcher was blind to experimental condition and/or the study hypothesis during data collection.* |
| Timing | *Indicate the start and stop dates of data collection. If there is a gap between collection periods, state the dates for each sample cohort.* |
| Data exclusions | *If no data were excluded from the analyses, state so OR if data were excluded, provide the exact number of exclusions and the rationale behind them, indicating whether exclusion criteria were pre-established.* |
| Non-participation | *State how many participants dropped out/declined participation and the reason(s) given OR provide response rate OR state that no participants dropped out/declined participation.* |
| Randomization | *If participants were not allocated into experimental groups, state so OR describe how participants were allocated to groups, and if allocation was not random, describe how covariates were controlled.* |

# Ecological, evolutionary & environmental sciences study design

All studies must disclose on these points even when the disclosure is negative.

| | |
|---|---|
| Study description | *Briefly describe the study. For quantitative data include treatment factors and interactions, design structure (e.g. factorial, nested, hierarchical), nature and number of experimental units and replicates.* |
| Research sample | *Describe the research sample (e.g. a group of tagged Passer domesticus, all Stenocereus thurberi within Organ Pipe Cactus National Monument), and provide a rationale for the sample choice. When relevant, describe the organism taxa, source, sex, age range and any manipulations. State what population the sample is meant to represent when applicable. For studies involving existing datasets, describe the data and its source.* |

| | |
|---|---|
| Sampling strategy | *Note the sampling procedure. Describe the statistical methods that were used to predetermine sample size OR if no sample-size calculation was performed, describe how sample sizes were chosen and provide a rationale for why these sample sizes are sufficient.* |
| Data collection | *Describe the data collection procedure, including who recorded the data and how.* |
| Timing and spatial scale | *Indicate the start and stop dates of data collection, noting the frequency and periodicity of sampling and providing a rationale for these choices. If there is a gap between collection periods, state the dates for each sample cohort. Specify the spatial scale from which the data are taken* |
| Data exclusions | *If no data were excluded from the analyses, state so OR if data were excluded, describe the exclusions and the rationale behind them, indicating whether exclusion criteria were pre-established.* |
| Reproducibility | *Describe the measures taken to verify the reproducibility of experimental findings. For each experiment, note whether any attempts to repeat the experiment failed OR state that all attempts to repeat the experiment were successful.* |
| Randomization | *Describe how samples/organisms/participants were allocated into groups. If allocation was not random, describe how covariates were controlled. If this is not relevant to your study, explain why.* |
| Blinding | *Describe the extent of blinding used during data acquisition and analysis. If blinding was not possible, describe why OR explain why blinding was not relevant to your study.* |

Did the study involve field work? ☐ Yes ☐ No

## Field work, collection and transport

| | |
|---|---|
| Field conditions | *Describe the study conditions for field work, providing relevant parameters (e.g. temperature, rainfall).* |
| Location | *State the location of the sampling or experiment, providing relevant parameters (e.g. latitude and longitude, elevation, water depth).* |
| Access & import/export | *Describe the efforts you have made to access habitats and to collect and import/export your samples in a responsible manner and in compliance with local, national and international laws, noting any permits that were obtained (give the name of the issuing authority, the date of issue, and any identifying information).* |
| Disturbance | *Describe any disturbance caused by the study and how it was minimized.* |

# Reporting for specific materials, systems and methods

We require information from authors about some types of materials, experimental systems and methods used in many studies. Here, indicate whether each material, system or method listed is relevant to your study. If you are not sure if a list item applies to your research, read the appropriate section before selecting a response.

### Materials & experimental systems

| n/a | Involved in the study |
|---|---|
| ☒ | ☐ Antibodies |
| ☐ | ☒ Eukaryotic cell lines |
| ☒ | ☐ Palaeontology and archaeology |
| ☒ | ☐ Animals and other organisms |
| ☒ | ☐ Human research participants |
| ☒ | ☐ Clinical data |
| ☒ | ☐ Dual use research of concern |

### Methods

| n/a | Involved in the study |
|---|---|
| ☒ | ☐ ChIP-seq |
| ☒ | ☐ Flow cytometry |
| ☒ | ☐ MRI-based neuroimaging |

## Antibodies

| | |
|---|---|
| Antibodies used | *Describe all antibodies used in the study; as applicable, provide supplier name, catalog number, clone name, and lot number.* |
| Validation | *Describe the validation of each primary antibody for the species and application, noting any validation statements on the manufacturer's website, relevant citations, antibody profiles in online databases, or data provided in the manuscript.* |

## Eukaryotic cell lines

Policy information about cell lines

| | |
|---|---|
| Cell line source(s) | HEK293S GnTI- were obtained from ATCC (identifier: CRL-3022), Sf9 were obtained from Thermo Fischer (identifier: 11496015) |
| Authentication | Cell lines were not authenticated. |

| Mycoplasma contamination | Cell lines were periodically tested for Mycoplasma contamination. |
| Commonly misidentified lines<br>(See ICLAC register) | No commonly misidentified cell lines were used in this study. |

# Palaeontology and Archaeology

| Specimen provenance | *Provide provenance information for specimens and describe permits that were obtained for the work (including the name of the issuing authority, the date of issue, and any identifying information). Permits should encompass collection and, where applicable, export.* |
| Specimen deposition | *Indicate where the specimens have been deposited to permit free access by other researchers.* |
| Dating methods | *If new dates are provided, describe how they were obtained (e.g. collection, storage, sample pretreatment and measurement), where they were obtained (i.e. lab name), the calibration program and the protocol for quality assurance OR state that no new dates are provided.* |

☐ Tick this box to confirm that the raw and calibrated dates are available in the paper or in Supplementary Information.

| Ethics oversight | *Identify the organization(s) that approved or provided guidance on the study protocol, OR state that no ethical approval or guidance was required and explain why not.* |

Note that full information on the approval of the study protocol must also be provided in the manuscript.

# Animals and other organisms

Policy information about studies involving animals; ARRIVE guidelines recommended for reporting animal research

| Laboratory animals | *For laboratory animals, report species, strain, sex and age OR state that the study did not involve laboratory animals.* |
| Wild animals | *Provide details on animals observed in or captured in the field; report species, sex and age where possible. Describe how animals were caught and transported and what happened to captive animals after the study (if killed, explain why and describe method; if released, say where and when) OR state that the study did not involve wild animals.* |
| Field-collected samples | *For laboratory work with field-collected samples, describe all relevant parameters such as housing, maintenance, temperature, photoperiod and end-of-experiment protocol OR state that the study did not involve samples collected from the field.* |
| Ethics oversight | *Identify the organization(s) that approved or provided guidance on the study protocol, OR state that no ethical approval or guidance was required and explain why not.* |

Note that full information on the approval of the study protocol must also be provided in the manuscript.

# Human research participants

Policy information about studies involving human research participants

| Population characteristics | *Describe the covariate-relevant population characteristics of the human research participants (e.g. age, gender, genotypic information, past and current diagnosis and treatment categories). If you filled out the behavioural & social sciences study design questions and have nothing to add here, write "See above."* |
| Recruitment | *Describe how participants were recruited. Outline any potential self-selection bias or other biases that may be present and how these are likely to impact results.* |
| Ethics oversight | *Identify the organization(s) that approved the study protocol.* |

Note that full information on the approval of the study protocol must also be provided in the manuscript.

# Clinical data

Policy information about clinical studies
All manuscripts should comply with the ICMJE guidelines for publication of clinical research and a completed CONSORT checklist must be included with all submissions.

| Clinical trial registration | *Provide the trial registration number from ClinicalTrials.gov or an equivalent agency.* |
| Study protocol | *Note where the full trial protocol can be accessed OR if not available, explain why.* |
| Data collection | *Describe the settings and locales of data collection, noting the time periods of recruitment and data collection.* |
| Outcomes | *Describe how you pre-defined primary and secondary outcome measures and how you assessed these measures.* |

# Dual use research of concern

Policy information about dual use research of concern

## Hazards

Could the accidental, deliberate or reckless misuse of agents or technologies generated in the work, or the application of information presented in the manuscript, pose a threat to:

No | Yes

☒ ☐ Public health

☒ ☐ National security

☒ ☐ Crops and/or livestock

☒ ☐ Ecosystems

☒ ☐ Any other significant area

## Experiments of concern

Does the work involve any of these experiments of concern:

No | Yes

☒ ☐ Demonstrate how to render a vaccine ineffective

☒ ☐ Confer resistance to therapeutically useful antibiotics or antiviral agents

☒ ☐ Enhance the virulence of a pathogen or render a nonpathogen virulent

☒ ☐ Increase transmissibility of a pathogen

☒ ☐ Alter the host range of a pathogen

☒ ☐ Enable evasion of diagnostic/detection modalities

☒ ☐ Enable the weaponization of a biological agent or toxin

☒ ☐ Any other potentially harmful combination of experiments and agents

# ChIP-seq

## Data deposition

☐ Confirm that both raw and final processed data have been deposited in a public database such as GEO.

☐ Confirm that you have deposited or provided access to graph files (e.g. BED files) for the called peaks.

Data access links
*May remain private before publication.*
> *For "Initial submission" or "Revised version" documents, provide reviewer access links. For your "Final submission" document, provide a link to the deposited data.*

Files in database submission
> *Provide a list of all files available in the database submission.*

Genome browser session
(e.g. UCSC)
> *Provide a link to an anonymized genome browser session for "Initial submission" and "Revised version" documents only, to enable peer review. Write "no longer applicable" for "Final submission" documents.*

## Methodology

Replicates
> *Describe the experimental replicates, specifying number, type and replicate agreement.*

Sequencing depth
> *Describe the sequencing depth for each experiment, providing the total number of reads, uniquely mapped reads, length of reads and whether they were paired- or single-end.*

Antibodies
> *Describe the antibodies used for the ChIP-seq experiments; as applicable, provide supplier name, catalog number, clone name, and lot number.*

Peak calling parameters
> *Specify the command line program and parameters used for read mapping and peak calling, including the ChIP, control and index files used.*

Data quality
> *Describe the methods used to ensure data quality in full detail, including how many peaks are at FDR 5% and above 5-fold enrichment.*

Software
> *Describe the software used to collect and analyze the ChIP-seq data. For custom code that has been deposited into a community repository, provide accession details.*

# Flow Cytometry

## Plots

Confirm that:

☐ The axis labels state the marker and fluorochrome used (e.g. CD4-FITC).

☐ The axis scales are clearly visible. Include numbers along axes only for bottom left plot of group (a 'group' is an analysis of identical markers).

☐ All plots are contour plots with outliers or pseudocolor plots.

☐ A numerical value for number of cells or percentage (with statistics) is provided.

## Methodology

| | |
|---|---|
| Sample preparation | *Describe the sample preparation, detailing the biological source of the cells and any tissue processing steps used.* |
| Instrument | *Identify the instrument used for data collection, specifying make and model number.* |
| Software | *Describe the software used to collect and analyze the flow cytometry data. For custom code that has been deposited into a community repository, provide accession details.* |
| Cell population abundance | *Describe the abundance of the relevant cell populations within post-sort fractions, providing details on the purity of the samples and how it was determined.* |
| Gating strategy | *Describe the gating strategy used for all relevant experiments, specifying the preliminary FSC/SSC gates of the starting cell population, indicating where boundaries between "positive" and "negative" staining cell populations are defined.* |

☐ Tick this box to confirm that a figure exemplifying the gating strategy is provided in the Supplementary Information.

# Magnetic resonance imaging

## Experimental design

| | |
|---|---|
| Design type | *Indicate task or resting state; event-related or block design.* |
| Design specifications | *Specify the number of blocks, trials or experimental units per session and/or subject, and specify the length of each trial or block (if trials are blocked) and interval between trials.* |
| Behavioral performance measures | *State number and/or type of variables recorded (e.g. correct button press, response time) and what statistics were used to establish that the subjects were performing the task as expected (e.g. mean, range, and/or standard deviation across subjects).* |

## Acquisition

| | |
|---|---|
| Imaging type(s) | *Specify: functional, structural, diffusion, perfusion.* |
| Field strength | *Specify in Tesla* |
| Sequence & imaging parameters | *Specify the pulse sequence type (gradient echo, spin echo, etc.), imaging type (EPI, spiral, etc.), field of view, matrix size, slice thickness, orientation and TE/TR/flip angle.* |
| Area of acquisition | *State whether a whole brain scan was used OR define the area of acquisition, describing how the region was determined.* |

Diffusion MRI     ☐ Used          ☐ Not used

## Preprocessing

| | |
|---|---|
| Preprocessing software | *Provide detail on software version and revision number and on specific parameters (model/functions, brain extraction, segmentation, smoothing kernel size, etc.).* |
| Normalization | *If data were normalized/standardized, describe the approach(es): specify linear or non-linear and define image types used for transformation OR indicate that data were not normalized and explain rationale for lack of normalization.* |
| Normalization template | *Describe the template used for normalization/transformation, specifying subject space or group standardized space (e.g. original Talairach, MNI305, ICBM152) OR indicate that the data were not normalized.* |
| Noise and artifact removal | *Describe your procedure(s) for artifact and structured noise removal, specifying motion parameters, tissue signals and physiological signals (heart rate, respiration).* |

| Volume censoring | *Define your software and/or method and criteria for volume censoring, and state the extent of such censoring.* |
|---|---|

## Statistical modeling & inference

| Model type and settings | *Specify type (mass univariate, multivariate, RSA, predictive, etc.) and describe essential details of the model at the first and second levels (e.g. fixed, random or mixed effects; drift or auto-correlation).* |
|---|---|
| Effect(s) tested | *Define precise effect in terms of the task or stimulus conditions instead of psychological concepts and indicate whether ANOVA or factorial designs were used.* |

Specify type of analysis: ☐ Whole brain  ☐ ROI-based  ☐ Both

| Statistic type for inference (See Eklund et al. 2016) | *Specify voxel-wise or cluster-wise and report all relevant parameters for cluster-wise methods.* |
|---|---|
| Correction | *Describe the type of correction and how it is obtained for multiple comparisons (e.g. FWE, FDR, permutation or Monte Carlo).* |

## Models & analysis

n/a | Involved in the study
☐ | ☐ Functional and/or effective connectivity
☐ | ☐ Graph analysis
☐ | ☐ Multivariate modeling or predictive analysis

| Functional and/or effective connectivity | *Report the measures of dependence used and the model details (e.g. Pearson correlation, partial correlation, mutual information).* |
|---|---|
| Graph analysis | *Report the dependent variable and connectivity measure, specifying weighted graph or binarized graph, subject- or group-level, and the global and/or node summaries used (e.g. clustering coefficient, efficiency, etc.).* |
| Multivariate modeling and predictive analysis | *Specify independent variables, features extraction and dimension reduction, model, training and evaluation metrics.* |

