## [Peer Review File. · Nature Structural & Molecular Biology]

Peer Review Information

Journal: Nature Structural and Molecular Biology

Manuscript Title: Structure of CRL7FBXW8 reveals coupling with CUL1-RBX1/ROC1 for multi-cullin-RING E3-catalyzed ubiquitin ligation

Corresponding author name(s): Professor Brenda Schulman

Reviewer Comments & Decisions:

Decision Letter, initial version:

24th Feb 2022

Dear Dr. Schulman,

Thank you again for submitting your manuscript "Structure of CRL7FBXW8 reveals coupling with CUL1-RBX1/ROC1 for multi-cullin-RING E3-catalyzed ubiquitin ligation". I apologize for the delay in responding, which resulted from the difficulty in obtaining suitable referee reports. Nevertheless, we now have comments (below) from the 2 reviewers who evaluated your paper. In light of those reports, we remain interested in your study and would like to see your response to the comments of the referees, in the form of a revised manuscript.

Please be sure to address/respond to all concerns of the referees in full in a point-by-point response and highlight all changes in the revised manuscript text file. If you have comments that are intended for editors only, please include those in a separate cover letter.

We expect to see your revised manuscript within 6 weeks. If you cannot send it within this time, please contact us to discuss an extension; we would still consider your revision, provided that no similar work has been accepted for publication at NSMB or published elsewhere.

Reporting Summary:

When submitting the revised version of your manuscript, please pay close attention to our [href="https://www.nature.com/nature-research/editorial-policies/image-integrity">Digital Image Integrity Guidelines. and to the following points below:](https://www.nature.com/nature-research/editorial-policies/image-integrity)

Please note that all key data shown in the main figures as cropped gels or blots should be presented in uncropped form, with molecular weight markers. These data can be aggregated into a single supplementary figure item. While these data can be displayed in a relatively informal style, they must refer back to the relevant figures. These data should be submitted with the final revision, as source data, prior to acceptance, but you may want to start putting it together at this point.

Data availability: this journal strongly supports public availability of data. All data used in accepted papers should be available via a public data repository, or alternatively, as Supplementary Information. If data can only be shared on request, please explain why in your Data Availability Statement, and also in the correspondence with your editor. Please note that for some data types, deposition in a public repository is mandatory - more information on our data deposition policies and available repositories can be found below:

<https://www.nature.com/nature-research/editorial-policies/reporting-standards#availability-of-data>

We require deposition of coordinates (and, in the case of crystal structures, structure factors) into the Protein Data Bank with the designation of immediate release upon publication (HPUB). Electron microscopy-derived density maps and coordinate data must be deposited in EMDB and released upon publication. Deposition and immediate release of NMR chemical shift assignments are highly

encouraged. Deposition of deep sequencing and microarray data is mandatory, and the datasets must be released prior to or upon publication. To avoid delays in publication, dataset accession numbers must be supplied with the final accepted manuscript and appropriate release dates must be indicated at the galley proof stage.

Nature Structural & Molecular Biology is committed to improving transparency in authorship. As part of our efforts in this direction, we are now requesting that all authors identified as 'corresponding author' on published papers create and link their Open Researcher and Contributor Identifier (ORCID) with their account on the Manuscript Tracking System (MTS), prior to acceptance. This applies to primary research papers only. ORCID helps the scientific community achieve unambiguous attribution of all scholarly contributions. You can create and link your ORCID from the home page of the MTS by clicking on 'Modify my Springer Nature account'. For more information please visit please visit www.springernature.com/orcid.

[REDACTED]

Sincerely,
Sara

Sara Osman, Ph.D.
Associate Editor
Nature Structural & Molecular Biology

Referee expertise:

Referee #1: CryoEM, Ubiquitylation

Referee #2: Structural biology, Biochemistry, Ubiquitylation

Reviewers' Comments:

Reviewer #1:

Remarks to the Author:

The manuscript by Hopf et al. describes the cryo-EM structure of CRL7FBXW8, an important but less characterized member of Cullin-Ring ubiquitin ligases (CRLs) essential for mammalian development. The structure is surprisingly different from that of canonical CRL complexes such as SCF-Skp2 and SCF- β -TrCP1 in that Skp1 is not engaged with the Cullin protein (Cul7). Instead, the WD40 domain of the F-box protein FBXW8 is involved in extensive interactions with Cul7. This new structure updates the current knowledge of CRLs by adding a new T-shaped complex to the previously reported C or O-shaped CRLs. Further, based on the complex structure which has an exposed Skp1 and a series of in vitro characterizations, the authors solved the puzzle of how CRL7-FBXW8 could interact with Cul1 in an FBXW8-dependent manner, providing a new paradigm of E3-E3 interaction. Finally, the authors successfully reconstituted the ubiquitin ligase activity of CRL7-FBXW8 in the presence of neddylated CRL1 and the substrate TP53, which will serve as a foundation for future mechanistic studies.

Overall, the paper is clearly written, with excellent displays of structural figures and interpretations of experimental data. The discoveries are novel and the conclusions are solid. The results presented are of interest to many researchers since multiple mutations of Cul7 have been found in 3M Syndrome, a rare but severe autosomal recessive disorder that affects children. Accordingly, it is rather important to elucidate the structural mechanism in order to understand the disease etiology. This paper certainly opens the door for future investigations.

There are no major concerns. Several questions the authors could clarify to further strengthen the paper:

1. Where might be the CPH domain located and how could a CPH engaged TP53 get close to the E2~Ub engaged by Cul1? From Fig. 3a, it seems that TP53 would be located to the left of the WD40 domain of FBXW8, which would prevent it from accessing the E2~Ub that is supposed to be on the right of WD40.
2. The authors argued that Cul7 engaged RBX1 would not engage E2~Ub based on the structural analyses and the absence of auto-ubiquitination of Cul7. Is there experimental evidence (e.g., binding assays) to support the lack of direct interactions with E2s? Would Cul7 engage an uncharged E2? If so, how would it fit into the model depicted in Fig. 5c?
3. Following question2, what would be the function of RBX in Cul7 complex if it is not engaging any E2? Would a super-complex as depicted in Fig. 5c but with only one RBX (engaged with Cul1) work as well?

The authors are not expected to fully address the above questions, which is beyond the scope of this paper. However, it would be great if the authors could provide further insights and thoughts.

Other minor points and suggestions are listed below:

1. In the first paragraph of Results, RMSD values are mentioned. Are these backbone RMSD or full-structure RMSD?
2. Are all the cryo-EM maps shown in the paper produced by DeepEMhancer? Specifically, the authors pointed to unassigned densities at the N- and C-terminal regions of Cul7 (Extended Data Fig. 3). Are these densities resolved better in the unsharpened map?
3. In the end of the second paragraph after “Unique cullin-F-box protein assembly”, the citation of Extended Data Fig. 4d should be Extended Data Fig. 5b.
4. Is the additional C-terminal helix of FBXW8 the determinant of the unique binding to Cul7? Why other WD40 domains of F-box proteins cannot bind to Cul7?
5. In the first paragraph after “Unique Cul7-Rbx1 assembly with FBXW8-Skp1 allows direct binding to Cul1-Rbx1”, CR1 deleted Cul1-Rbx1 is not shown in Fig. 5b (only in Extended Data Fig. 7a).
6. In the second paragraph after “CRL7-FBXW8 can serve as a substrate receptor for neddylylated CRL1 in vitro”, Fig. 5b should be cited in all the places referring to “lane”.
7. The structural model has 7.2% poor rotamers, which can be improved by rotamer analysis in Coot. The other statistics are fine.

Reviewer #2:

Remarks to the Author:

The multi-component cullin-RING ligases are the largest family of ubiquitin ligases in mammalian cells and have enormous compositional variety due to a “mix-and-match” system of 9 different cullin bodies with various RING domains and substrate adaptors. While a number of high-resolution structures of distinct cullin complexes were determined, no structural information was available on the CRL assembly CRL7FBW8. Previous studies suggested that CUL7-based complexes have atypical features, however, the structural and mechanistic basis and functional consequences of these features remained elusive. This manuscript reports a cryo-EM structure of CRL7FBW8, which – together with in vitro binding and activity assays - delineates the special architecture and non-canonical interactions of this complex compared to other CRLs. The structure is discussed in exquisite detail and is compared to other available structures with great proficiency. The biochemical work is also very well designed, meaningful, and clear. The data converge into a model in which CRL7FBW8

per se is catalytically inactive, but functions as a substrate receptor together with an active, neddylated CUL1-RBX1 module. This is a highly interesting scenario, which adds to the emerging notion that ligases cooperate with each other, as exemplified by other recent work from the Schulman laboratory. Taken together, I highly recommend this excellent, original work for publication in NSMB.

I only have few suggestions on how to improve this manuscript:

- (1) In order to back up the idea that immunoprecipitated CUL7 (but not purified CUL7) is active in TP53 ubiquitination due to its binding to CUL1-RBX1, it would be good to perform such an IP and blot against a component of the CUL1-RBX1 complex or detect CUL1 in the immunoprecipitate by MS (unless this has been done before).
- (2) If practically feasible, I recommend that the authors include stained gels of the activity assays (e.g., in the Supplements) along with the fluorescence scans to visualize the input.
- (3) It would be helpful to state which protein classes OBSL1 and CCDC8 belong to when they are first mentioned (page 3).
- (4) It should be specified which type of RMSD is quoted when comparing the cryoEM structure with the AlphaFold model.

Author Rebuttal to Initial comments

General response to reviewers:

We thank the Reviewers for their encouraging comments and suggestions for improving our manuscript. Point-by-point responses are in blue.

Reviewer #1:

The manuscript by Hopf et al. describes the cryo-EM structure of CRL7FBXW8, an important but less characterized member of Cullin-Ring ubiquitin ligases (CRLs) essential for mammalian development. The structure is surprisingly different from that of canonical CRL complexes such as SCF-Skp2 and SCF- β -TrCP1 in that Skp1 is not engaged with the Cullin protein (Cul7). Instead, the WD40 domain of the F-box protein FBXW8 is involved in extensive interactions with Cul7. This new structure updates the current knowledge of CRLs by adding a new T-shaped complex to the previously reported C or O-shaped CRLs. Further, based on the complex structure which has an exposed Skp1 and a series of in vitro characterizations, the authors solved the puzzle of how CRL7-FBXW8 could interact with Cul1 in an FBXW8-dependent manner, providing a new paradigm of E3-E3 interaction. Finally, the authors successfully reconstituted the ubiquitin ligase activity of CRL7-FBXW8 in the presence of neddylated CUL1 and the substrate TP53, which will serve as a foundation for future mechanistic studies.

Overall, the paper is clearly written, with excellent displays of structural figures and interpretations of experimental data. The discoveries are novel and the conclusions are solid. The results presented are of interest to many researchers since multiple mutations of Cul7 have been found in 3M Syndrome, a rare but severe autosomal recessive disorder that affects children. Accordingly, it is

rather important to elucidate the structural mechanism in order to understand the disease etiology. This paper certainly opens the door for future investigations.

We thank the reviewer for such kind comments and enthusiasm for our study!

There are no major concerns. Several questions the authors could clarify to further strengthen the paper:

1. Where might be the CPH domain located and how could a CPH engaged TP53 get close to the E2~Ub engaged by Cul1? From Fig. 3a, it seems that TP53 would be located to the left of the WD40 domain of FBXW8, which would prevent it from accessing the E2~Ub that is supposed to be on the right of WD40.

Although we do not know the precise location of the CPH domain, several observations lead us to believe this domain is flexibly tethered to its connections within the ARM2 domain. First, in the cryo-EM data, we do not observe the domain itself, nor the upstream ≈ 45 residues nor the downstream ≈ 28 -residues connecting the CPH domain to the ARM2 domain. Second, these linker sequences are predicted to be intrinsically disordered. Third, comparing the AlphaFold models of the highly similar human, orangutan, rat and mouse CUL7, the CPH domain is in variable locations relative to the remainder of CUL7 (notably the rest of CUL7 models superimpose). ≈ 45 - and ≈ 28 -residue flexible linkers would be long enough to span the ≈ 70 and ≈ 50 Å unencumbered distances between the ARM2 domain and the modeled active site of the neddylated CUL1-RBX1-bound UBE2D, although we presume that some of the distance would be spanned by the CPH domain itself (which is ≈ 25 Å across) and the p53 tetramer.

To address this question in the revised manuscript, we added the following text and panel in Extended Data Figure 8:

“The CPH domain is connected to the ARM2 domain by ≈ 45 - and ≈ 28 -residue long linker sequences, which are presumably flexible given that this domain is not observed in the cryo-EM map. In principle, the CPH domain - which itself spans ≈ 25 Å - could extend over a wide radius relative to the rest of the complex to deliver TP53 to the ubiquitination active site (Extended Data Fig. 8a).”

a Superposition of CRL1 ^{β -TRCP} (6TTU) onto SKP1 of CRL7^{FBXW8}

2. The authors argued that Cul7 engaged RBX1 would not engage E2~Ub based on the structural analyses and the absence of auto-ubiquitination of Cul7. Is there experimental evidence (e.g., binding assays) to support the lack of direct interactions with E2s? Would Cul7 engage an uncharged E2? If so, how would it fit into the model depicted in Fig. 5c?

We and others have found that E2 binding to the RBX1 RING domain is extremely weak - for the best binder (UBE2R-family E2s) this has been detected only by NMR, with K_D estimated to be at least in the millimolar range. We were unable to achieve suitably high concentrations of either CUL1-RBX1 as a control, or CUL7-RBX1 to perform binding assays. We and others have found that canonical CRLs recruit ubiquitin-loaded E2s via multivalent interactions. Ubiquitin-linked to an E2 active site was estimated by Spratt and co-workers to increase the affinity for the RBX1 RING domain 50-fold. E2s in the UBE2R family have a C-terminal acidic extension that binds to a cullin region termed "basic canyon" (notably CUL7 does not have this basic canyon). Also, the cullin-linked NEDD8 further contributes to canonical CRL recruitment of ubiquitin carrying enzymes. Due to these challenges, we turned toward autoubiquitination as an assay for intrinsic activity with UBE2D-family and other E2s.

Nonetheless, the reviewer question also prompted us to perform di-ubiquitin synthesis assays testing if CUL7 could activate ubiquitin chain formation by E2s (UBE2G1 and UBE2R1) that mediate this activity with canonical CRLs. We also did not observe such activity for CUL7-RBX1, even though CUL1-RBX1 robustly activated di-ubiquitin synthesis under the same conditions. These data are in Extended Data Figure 6g of the revision.

We also addressed the reviewer question in two sections of the text, as follows:

"Canonical CRLs exhibit a wide range of E3 ligase activities, in partnership with various NEDD8- and ubiquitin-linked carrying enzymes. These activities depend on multivalent interactions, at minimum involving the RBX RING domain, an E2, and its active site-linked NEDD8 or ubiquitin, and also other contacts, for example between a cullin-linked NEDD8 and a ubiquitin-carrying enzyme^{41,42,45}."

"In complexes with neddylated canonical cullins, RBX1's RING domain can also promote ubiquitin chain formation by UBE2G1 and by UBE2R-family E2s. RBX1 associated with unneddylated canonical cullins can also promote such polyubiquitination with UBE2R1, which is stimulated by interactions between an acidic C-terminal region of UBE2R1 and a basic canyon on canonical cullins that is strikingly lacking in CUL7 (Extended Data Fig. 6f). Importantly, these activities can be assayed even in the absence of a substrate, by monitoring transfer of fluorescent ubiquitin from a pre-formed E2~ubiquitin intermediate to an unlabeled acceptor ubiquitin ("~" here refers to thioester linkage between E2 catalytic cysteine and ubiquitin C-terminus). However, CUL7-RBX1 was inactive under conditions when RBX1 complexes with CUL1 promote di-ubiquitin synthesis (Extended Data Fig. 6g)."

3. Following question2, what would be the function of RBX in Cul7 complex if it is not engaging any E2? Would a super-complex as depicted in Fig. 5c but with only one RBX (engaged with Cul1) work as well?

Prior studies have shown that co-expression of RBX1 N-terminus is required for proper folding of canonical CRLs, because its N-terminal β 3-strand contributes to an intermolecular β 3-sheet in a so-called "cullin/RBX" or "C/R" domain. We also had to coexpress RBX1 to obtain well-behaved CUL7.

Indeed, the intermolecular CUL7-RBX1 C/R domain superimposes on that of canonical CRLs (and on the corresponding intermolecular domain between APC2 and APC11 within the APC/C).

To clarify this, we describe the C/R domain in the Results as follows:

RBX1's N-terminal region forms a β -strand, which is inserted in CUL7's α / β region. Together, these CUL7 and RBX1 elements form an intermolecular cullin/RBX (C/R) domain that requires both proteins for proper folding.

We also clarify that RBX1 contributes to proper folding during in the Discussion when asking:

But does RBX1 play additional functions beyond promoting proper CUL7 folding in the context of other binding partners?

The authors are not expected to fully address the above questions, which is beyond the scope of this paper. However, it would be great if the authors could provide further insights and thoughts.

Other minor points and suggestions are listed below:

1. In the first paragraph of Results, RMSD values are mentioned. Are these backbone RMSD or full-structure RMSD?

We thank both reviewers for pointing this out. The values are calculated over Ccts. We have edited the text as follows:

“Although AlphaFold was released subsequent to structure determination, it is noteworthy that most of the individual domains in CUL7 and FBXW8 superimpose with those predicted with Cct-RMSD values of $< 1.3 \text{ \AA}$ and $< 0.8 \text{ \AA}$, respectively (Extended Data Fig. 2)³⁹.”

2. Are all the cryo-EM maps shown in the paper produced by DeepEMhancer? Specifically, the authors pointed to unassigned densities at the N- and C-terminal regions of Cul7 (Extended Data Fig. 3). Are these densities resolved better in the unsharpened map?

The maps in the original figures were produced by DeepEmhancer. The unassigned densities that we presume correspond to the N- and C-terminal regions of CUL7 are not better resolved in the unsharpened map. However, the additional density covering RBX1's RING domain is clearly visible in the new, lower resolution, map of the complex between CUL7^{FBXW8} and CUL1's N-terminal region. It is possible that this interaction is sufficiently conformationally heterogeneous to become less visible at higher resolution, although the low resolution precludes assigning sequence to this density. To bolster this point in the revised manuscript, we show a close-up in Extended Data Fig. 8b.

Not included in the manuscript, but to aid the reviewers, we include here views of the density covering RBX1 at different contour levels in unsharpened and sharpened maps of CRL7^{FBXW8} in comparison with the map of CRL7^{FBXW8}-CUL1 N-terminal region.

3. In the end of the second paragraph after “Unique cullin-F-box protein assembly”, the citation of Extended Data Fig. 4d should be Extended Data Fig. 5b.

We thank the reviewer for catching this. We fixed the figure citation in the revised manuscript.

4. Is the additional C-terminal helix of FBXW8 the determinant of the unique binding to Cul7? Why other WD40 domains of F-box proteins cannot bind to Cul7?

CUL7 binds FBXW8 through an elaborate multipart interface. All parts of the interface involve sequences unique to FBXW8 amongst F-box proteins (the sequence between the F-box and stalk helix, the stalk helix, insertions in the WD40 propeller, sequences within the propeller unique to FBXW8, and the C-terminal extension). In addition, amongst human F-box proteins with WD40 domains, only FBXW7 and FBXW8 seem to have 8-bladed propellers (the others either have or are

predicted to have fewer blades). The CUL7-binding sequences are not conserved in FBXW7, and we now show this in new figure panels (Extended Data Fig. 5b-c).

We also revised the text after describing the CUL7-FBXW8 interactions as follows:

"The atypical assembly explains why CUL7 uniquely binds FBXW8 but no other F-box proteins: the interactions are with domains, domain insertions, and sequences unique to FBXW8, and even those with the WD40 domain involve residues that are not conserved even with its closest homolog FBXW7 (Extended Data Fig. 5b-c)."

c conserved residues between FBXW8 and FBXW7

5. In the first paragraph after “Unique Cul7-Rbx1 assembly with FBXW8-Skp1 allows direct binding to Cul1-Rbx1”, CR1 deleted Cul1-Rbx1 is not shown in Fig. 5b (only in Extended Data Fig. 7a).

We thank the reviewer for catching this. We fixed the figure citation in the revised manuscript.

6. In the second paragraph after “CUL7-FBXW8 can serve as a substrate receptor for neddylated CUL1 in vitro”, Fig. 5b should be cited in all the places referring to “lane”.

We thank the reviewer for catching this. We fixed the figure callouts in the revised manuscript.

7. The structural model has 7.2% poor rotamers, which can be improved by rotamer analysis in Coot. The other statistics are fine.

We thank the reviewer for catching this. The final revised model has 0.2% poor rotamers. The updated statistics are provided in Table 1. We also provide a PDB validation report.

Reviewer #2:

The multi-component cullin-RING ligases are the largest family of ubiquitin ligases in mammalian cells and have enormous compositional variety due to a "mix-and-match" system of 9 different cullin bodies with various RING domains and substrate adaptors. While a number of high-resolution structures of distinct cullin complexes were determined, no structural information was available on the CRL assembly CRL7FBW8. Previous studies suggested that CUL7-based complexes have atypical features, however, the structural and mechanistic basis and functional consequences of these features remained elusive. This manuscript reports a cryo-EM structure of CRL7FBW8, which – together with in vitro binding and activity assays - delineates the special architecture and non-canonical interactions of this complex compared to other CRLs. The structure is discussed in exquisite detail and is compared to other available structures with great proficiency. The biochemical work is also very well designed, meaningful, and clear. The data converge into a model in which CRL7FBW8 per se is catalytically inactive, but functions as a substrate receptor together with an active, neddylated CUL1-RBX1 module. This is a highly interesting scenario, which adds to the emerging notion that ligases cooperate with each other, as exemplified by other recent work from the Schulman laboratory. Taken together, I highly recommend this excellent, original work for publication in NSMB.

We thank the reviewer for such kind comments and enthusiasm for our study! I only have few suggestions on how to improve this manuscript:

(1) In order to back up the idea that immunoprecipitated CUL7 (but not purified CUL7) is active in TP53 ubiquitination due to its binding to CUL1-RBX1, it would be good to perform such an IP and blot against a component of the CUL1-RBX1 complex or detect CUL1 in the immunoprecipitate by MS (unless this has been done before).

This was first demonstrated by Nakayama and colleagues (Tsunematsu et al., 2006). We now describe their experiment in more detail in the Introduction:

"...in vivo, CRL7^{FBW8} curiously associates with CUL1-RBX1. This interaction requires FBW8: in wild-type MEFs, exogenously expressed Flag-tagged CUL1 immunoprecipitates endogenous CUL7 and Flag-tagged CUL7 immunoprecipitates endogenous CUL1. However, the CUL7-CUL1 interactions were not observed in FBW8 null cells²²."

We provide here the data from their paper, as this laid a foundation for our own study:

FIG. 7. Formation of a Cul1-Cul7 complex linked by Fbxw8. *Fbxw8*^{+/+} or *Fbxw8*^{-/-} MEFs were infected with recombinant retroviruses encoding 3× FLAG (3F)-tagged Cul1 (A) or Cul7 (B), and lysates of the infected cells were subjected to immunoprecipitation with anti-FLAG. The resulting precipitates as well as the original cell lysates were then subjected to immunoblot analysis with the indicated antibodies. Cells expressing 3× FLAG-Cul2 were also studied as a negative control. IB, immunoblot; IP, immunoprecipitation.

To further strengthen our conclusions, we were able to obtain a 4.6 Å resolution cryo-EM map of CUL7^{FBXW8} in complex with the N-terminal region of CUL1 (residues 1-410, encompassing CUL1's CR1, CR2, and CR3 domains). The map matches our proposed model and we now include this in the revision as follows:

“Indeed, a cryo-EM map of a complex between CUL7^{FBXW8} and the N-terminal region of CUL1 (comprising the CR1, CR2, and CR3 domains), refined to a resolution of 4.6 Å visualized the

interactions: SKP1-FBXW8 within the CUL7FBXW8 complex binds CUL1's CR1 domain in a canonical manner (Fig. 5b).”

(2) If practically feasible, I recommend that the authors include stained gels of the activity assays (e.g., in the Supplements) along with the fluorescence scans to visualize the input.

We appreciate this suggestion. Where possible, we have added the Coomassie stained gels to the raw image data file. These include all the assays in the main figures.

(3) It would be helpful to state which protein classes OBSL1 and CCDC8 belong to when they are first mentioned (page 3).

We added these details to the Introduction:

"The CCDC8 protein was named for its predicted coiled-coil domain, is thought to contain several intrinsically unstructured regions that mediate protein interactions, and localizes at the plasma membrane^{19,25–30}. OBSL1 is a putative cytoskeletal adaptor protein with multiple immunoglobulin-like domains and a fibronectin type-3 domain³¹."

(4) It should be specified which type of RMSD is quoted when comparing the cryoEM structure with the AlphaFold model.

We thank both reviewers for pointing this out. The values are calculated over Cas. We have edited the text as follows:

"Although AlphaFold was released subsequent to structure determination, it is noteworthy that most of the individual domains in CUL7 and FBXW8 superimpose with those predicted with Ca-RMSD values of $< 1.3 \text{ \AA}$ and $< 0.8 \text{ \AA}$, respectively (Extended Data Fig. 2)³⁹."

Decision Letter, first revision:

24th Mar 2022

Dear Dr. Schulman,

Thank you for submitting your revised manuscript "Structure of CUL7FBXW8 reveals coupling with CUL1-RBX1/ROC1 for multi-cullin-RING E3-catalyzed ubiquitin ligation" (NSMB-A45750A). It has now been seen by the original referees and their comments are below. The reviewers find that the paper has improved in revision, and therefore we'll be happy in principle to publish it in Nature Structural & Molecular Biology, pending minor revisions to satisfy the referees' final requests and to comply with our editorial and formatting guidelines.

To facilitate our work at this stage, we would appreciate if you could send us the main text as a word file. Please make sure to copy the NSMB account (cc'ed above).

Sincerely,
Sara

Sara Osman, Ph.D.
Associate Editor
Nature Structural & Molecular Biology

Reviewer #1 (Remarks to the Author):

The authors have successfully addressed all my concerns. I enthusiastically support its publication in NSMB.

Reviewer #2 (Remarks to the Author):

Everything has been addressed superbly.

Final Decision Letter: